


# Multi-scale EO-based agricultural drought monitoring system for operative irrigation networks management

Chiara Corbari[1], Nicola Paciolla[1], Giada Restuccia[1], and Ahmad Al Bitar[2]

[1]Politecnico di Milano – Piazza Leonardo da Vinci, 32 - 20133 – Milan (Italy)
[2]CESBIO (Université de Toulouse, CNES, CNRS, IRD) – Toulouse (France)

**Correspondence:** Chiara Corbari (chiara.corbari@polimi.it)

**Abstract.** Drought prediction is crucial especially where the rainfall regime is irregular and agriculture is mainly based on irrigated crops, such as in Mediterranean countries. In this work, the main objective is to develop an EO-based agricultural drought monitoring system (ADMOS) for operative irrigation networks management at different spatial and temporal scales. Different levels of drought are identified based on an integrated indicator combining anomalies of rainfall, soil moisture, land surface temperature and vegetation indices, allowing to consider the different droughts types and their timing looking on the end-user's perspective. Multiple remote sensing data, which differ on sensing techniques, spatial and temporal resolutions and electromagnetic frequencies, are used for each anomaly computation. The analyses have been performed over two Irrigation Consortia in Italy (the Chiese and Capitanata ones), which differ for climate, irrigation volumes and techniques, and crop types. The obtained results show a negative correlation between cumulated ADMOS and the irrigation volumes in the Capitanata area, while in the Chiese Consortium a zero correlation is obtained with an almost constant amount of irrigation volumes provided to the crops every year independently from the drought condition. In both areas, crop yields seem to be almost uncorrelated to the drought index, as production is highly sustained by irrigation. Moreover, discrepancies on the anomalies sign is observed, especially when soil moisture is considered. The results also clearly show that asynchronies may exist especially between soil moisture anomalies and vegetation or land surface temperature anomalies.

## 1 Introduction

Drought is an intensifying hazard with a complex behavior affecting either the natural and anthropogenic environments as well as the socio-economic activities (Wood et al., 2015; Vicente-Serrano, 2006; Sheffield et al., 2009; Van Loon et al., 2014; Wilhite et al., 2007; Mann and Gleick, 2015). According to the Munich Re report (2020), between 2000 to 2019, the 35% of hazard events are related to droughts affecting more than 1.43 billion people: as the Australian 2000 event (van Dijk et al., 2013), the drought of 2010 in Russia (Spinoni et al., 2018) or of 2003 in central Europe (García-Herrera et al., 2010), the 2013–2014 California drought (Swain et al., 2014), the Chinese event of 2014 (Wang and He, 2015) and the southern Africa drought of 2015–2017 (Meza et al., 2021). In particular, strong negative impacts have been reported on the agricultural sector, mainly due to water shortages with reductions of crop productions and consequently economic losses (Ding et al., 2011). These droughts events are further projected to increase due to climate change and also increased land-atmosphere feedbacks due to changed



land/vegetation coverage and also irrigated areas Spinoni et al. (2018). Thus, agriculture, which is by far the largest water user with about 70% of total freshwater consumption (Alexandratos and Bruinsma, 2012), will be more and more impacted, even though irrigated areas cover only about 2% of the global land area (Food and of the United Nations , FAO). However, without irrigation, crops could never be grown in the deserts of California, Israel or even in South of Italy or Spain.

Therefore, an improvement in the understanding of the spatial and temporal variability of droughts is required to meet the
increasing effects of droughts (AghaKouchak et al., 2015).

Drought is generally defined as a temporal anomaly in comparison with average conditions over long time periods (Wood et al., 2015). Wilhite and Glantz (1985) identified four different types of droughts based on their impacts: i) meteorological as a precipitation deficiency, ii) agricultural as soil water availability shortage, iii) hydrological for low runoff and groundwater deficiency, iv) socioeconomic on the society. Each type of drought might evolve with different timing, intensity, duration
and spatial extent (Vicente-Serrano, 2006). To describe this complex phenomenon, a huge number of drought indices have been developed over the years to describe the different processes and for different applications, without however being able to identify a best drought index to describe the multifaceted nature of drought (Wanders et al., 2017). Therefore, the World Meteorological Organization (WMO) and Global Water Partnership (GWP) (2016) suggest to use multiple drought indices.

Among them, the Standardized Precipitation Index (SPI) is defined by the WMO as the standard index for analysing a me-
teorological drought, by fitting the precipitation data to a standardized normal probability distribution function and evaluating the number of standard deviations by which the anomaly deviates from the mean, over different time scales from 1 to 36 months (McKee et al., 1993). SPI advantages are mainly that it relies only on precipitation and its relatively simple computation (Kumar et al., 2016), while the key limitation is relative to the high sensitivity to the quality and time length of data used, and moreover it does not account for the atmospheric water demand. The Standardized Precipitation Evapotranspiration
Index (SPEI, Vicente-Serrano et al., 2010) and the Palmer Drought Severity Index (PDSI,  Sheffield et al., 2012) overcome this limitation by considering the evapotranspiration besides precipitation, but are limited by the non-comparability across regions, the use of a potential evapotranspiration and the inclusions of snow effects.

An alternative approach is to monitor soil moisture (SM) changes over large areas using satellite-based SM products (Sadri et al., 2018), while several satellite-based products exist based on sampling techniques in the microwave part of the electro-
magnetic spectrum with active (Sentinel-1, ASCAT) or passive (SMOS, SMAP, AMSR-E) technologies (Kerr et al., 2010; Bauer-Marschallinger et al., 2018). However, inconsistencies among the different products have been documented with correlations ranging from 0.48 to 0.89 among them and ground data (Paciolla et al., 2020; Cui et al., 2017; El Hajj et al., 2018). This might be mainly due to different sensing techniques, spatial resolutions (from 50 km of SMOS to 1 km of Sentinel-1), the sensing soil depth of few centimetres, which is not congruent with the hydrological active soil for plant root zone uptake.
Moreover, there are problems linked to the saturation of soil moisture retrieval algorithms for active radars (Giacomelli et al., 1995) and their ability to detect soil moisture over vegetated surfaces (Bindlish and Barros, 2001).

A less used approach for drought monitoring is the use of a Land Surface Temperature (LST) derived index, computed using the thermal infrared (TIR) spectra on-board a few numbers of sensors (MODIS, LANDSAT, Sentinel-3) (Jimenez-Munoz et al., 2014). The LST is widely used in energy or energy-water balance models to compute evapotranspiration (Corbari and


Mancini, 2022; Su, 2002; Kustas and Norman, 1999), while for longer periods analysis is almost unused mainly due to data unavailability during cloud conditions and the non-unique relationship between LST and SM (Price, 1980).

Several satellite vegetation indices might be used to monitor the effect of droughts directly on vegetations, as the NDVI index, which however might be limited in its use by other external factors (e.g. soil adjacency), or the normalized difference water index (NDWI) which accounts for the water leaf contents, of the fraction of absorbed photosynthetically active radiation

(fAPAR, Choudhury, 1987).

Moreover, soil moisture anomalies are sometimes asynchronous respect to vegetation ones (van Hateren et al., 1990), due to the delayed response of vegetation to water scarcity or vegetation anomalies not related to water availability, which can lead to a discrepancy between the definition of agricultural droughts and hence false drought identifications. To overcome this inconsistency, it is thus important to detect the different droughts on SM and vegetation.

Thus, for operative applications for supporting policymakers, water managers and stakeholders, a combination of indices is an optimal solution to detect the different droughts and their impacts. In the last few years, some operational drought monitoring and early warning systems have been developed at continental scales, by assembling a suite of independent indicators which detect the different type of droughts, to be able to provide water shortage trends and deviations from the long-term averages of mainly precipitation, streamflow, soil moisture and groundwater (Vogt et al., 2018; Sheffield et al., 2014). Among them, the

United States Drought Monitor (https://droughtmonitor.unl.edu), based on different indicators at weekly scale, or the Global Integrated Drought Monitoring and Prediction System (GIDMaPS, http://drought.eng.uci.edu), or the Combined Drought Indicator (CDI, Sepulcre-Canto et al., 2012) revised by Cammalleri et al. (2021) for considering the consistency of prolonged and interrupted drought events, which is used into the European Drought Observatory (EDO, https://edo.jrc.ec.europa.eu) for detecting European droughts. The CDI index is based on the combined use of SPI, remote sensing fAPAR and soil moisture

from a hydrological model, allowing to define three drought categories: watch with precipitation below normal, warning when the precipitation deficit is propagated in soil moisture deficit, alert when it also affects the vegetation.

In this framework, this work is answering to the question if it is possible to develop an EO-based agricultural drought monitoring system (ADMOS) for operative irrigation networks management. The monitoring system has been developed following the principle of the temporal evolution of the drought process, by firstly considering the meteorological and agricultural

droughts, vegetation water stress and drying. Different levels of drought are then assigned considering the severity computed from the combination of the different droughts' indicators, starting from the no drought condition till the extreme drought one. This drought monitoring system is developed identifying the relationships between the causes (lack of precipitation, high temperatures and radiations) and effects (soil moisture shortage, vegetation water stress and vegetation drying). Such a relationship gives the opportunity to provide a system for water stress and crop failure monitoring for activating crops protection

actions, as irrigation. The robustness of the developed ADMOS will be evaluated by assessing the effect of a drought event on irrigation volumes and crop yields at Irrigation Consortium Scale. The methodology is applied in two Italian cases studies: the Chiese and Capitanata Irrigation Consortia, which are located in the North and South of Italy, respectively. These areas mainly differ for climatic conditions, crops types, irrigation strategies and techniques. The analyses are performed over 20 years for the period from 2000 to 2019.




The main innovations rely on: i) the analyses of ADMOS which is usable for operative water management at regional to farm scale; ii) the direct comparison of the drought monitoring system on the effective used irrigation volumes and the crop yields at Irrigation Consortium Scale over different years to determine its robustness, iii) a method which is fully based on remote sensing data, differing on sensing techniques, spatial and temporal resolutions, to detect the full drought process development; iv) insertion of the LST anomaly to detect the plant water stress, thanks its role in the energy balance for latent and sensible heat fluxes partitioning.

## 2   Materials and Methods

### 2.1   The agricultural drought monitoring system: ADMOS

Following the principle of the temporal evolution of the drought process (Wilhite and Glantz, 1985), the meteorological drought occurs first (a period with less rainfall than average, according to the SPI index), and it generally precedes the agricultural drought, when crop production may be reduced due to soil moisture shortage. This shortage can further increase the drought severity, leading to an increase of crop surface temperature (observed by vegetation water stress indices) and finally to vegetation drying. Different levels of drought are then assigned considering the severity computed from the combination of the different droughts' indicators, starting from the no drought condition till the extreme drought one.

The ADMOS indicator is divided into 4 steps which are presentative of each drought step: 1) precipitation deficit is assessed using the standardized precipitation index (SPI) index, 2) soil moisture shortage is evaluated by the soil moisture anomaly (SMA), 3) vegetation drying is identified with a land surface temperature anomaly (LSTA) and finally 4) vegetation stress is estimated by a vegetation index anomaly (VISA). During the irrigation season (from April to October) irrigation is added to precipitation. The drought indicator can take the following values reported in Figure 1 according to the increased drought.

Anomalies are evaluated as the deviation of a parameters from a long-term mean value at monthly scale to remove the seasonality signal by computing:

$$MonthlyAnomaly = \frac{x_{im} - \overline{x}_m}{\overline{\sigma}_{Xm}} \qquad (1)$$

where $X_{im}$ is the variable value $i$-th day of the $m$ month, $\bar{X}_m$ and $\bar{\sigma}_{Xm}$ are respectively the mean and standard deviation calculated along each $m$ month of the year. The anomalies are computed at daily scale for soil moisture, land surface temperature and the vegetation indices.

The Standardized Precipitation Index (SPI, McKee et al., 1993) represents the probability that the location would have received at least the observed amount of precipitation over a time period. In this work, using daily precipitation values, the SPI on 1 month (SPI-1) is computed. Positive SPI values represent wet conditions; while negative SPI values represent dry conditions. The historic dataset is fitted to the gamma probability distribution, where the shape and scale parameters of the distribution are first fitted on the frequency distribution of the historical non-zero rainfall data using the approximation of Thom





| Index value | SPI-1 | SMA | LSTA | VISA | Condition |
|---|---|---|---|---|---|
| 1 | >0 | >0 | <0 | >0 | Surplus of water |
| 0.5 | >0 | >0 | <0 | <0 | Surplus of water |
| 0 | >0 | >0 | <0 | >0 | No drought |
| -0.5 | <0 & >1 | | | | Mild Drought |
| -1 | <-1 | <0 | | | |
| -1.5 | <0 | <0 & >-1 | | | Moderate Drought |
| -2 | <0 | <-1 | >0 | | |
| -2.5 | <0 | <0 | >0 & <1 | | Severe Drought |
| -3 | <0 | <0 | >1 | <0 | |
| -3.5 | <0 | <0 | >0 | <0 & >-1 | Extreme Drought |
| -4 | <0 | <0 | >0 | <-1 | |

**Figure 1.** Classification of ADMOS drought according to the anomalies' combinations

(1958). The cumulative probability is then computed based on the parameters of the gamma distribution and then transformed
into a normal distribution with zero mean by using the approximate conversion provided by Abramowitz and Stegun (1964).

### 2.1.1 Statistical Indices

To determine the differences among the different products of the different variables, the Root Mean Square Error (RMSE) is
used:

$$RMSE = \sqrt{\sum_{i=1}^{n} \frac{(X_i - Y_i)^2}{n}} \qquad (2)$$

Where $n$ is the sample size, $X_i$ and $Y_i$ are the two considered dataset. Then, the Pearson correlation coefficient ($r_{XY}$) is
also computed as a measure of the linear correlation between two variables. It has a value between +1 and -1, where 1 is total
positive linear correlation, 0 is no linear correlation, and -1 is total negative linear correlation. The coefficient is calculated as:

$$r_{XY} = \frac{\sum_{i=1}^{n}(x_i - \bar{x})(y_i - \bar{y})}{\sqrt{\sum_{i=1}^{n}(x_i - \bar{x})^2 \sum_{i=1}^{n}(y_i - \bar{y})^2}} \qquad (3)$$

Where $\bar{x}$, $\bar{y}$ are the samples means.





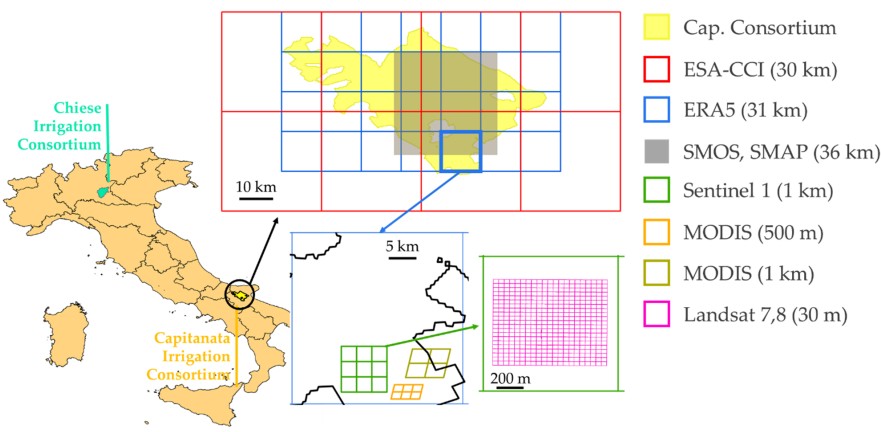

**Figure 2.** Location of the Chiese and Capitanata Irrigation Consortia within Italy, highlighting, in the example of the latter, the footprints of the satellite products of interest. Coarse-scale datasets (e.g., ESA-CCI products, ERA5, SMOS and SMAP) require a limited number of pixels to cover the entire Consortium area, while higher-resolution (500 m — 1 km) products (such as Sentinel 1 and MODIS) provide more local information. The finest-resolution dataset is Landsat (both 7 and 8), with a spatial resolution of 30 m which is below the characteristic field size and is thus able to provide intra-field spatial variability.

## 2.2 Case studies and data

The methodology is applied in two cases studies: the Chiese and Capitanata Irrigation Consortia, which are located in the North and South of Italy, respectively (Fig.2). These two areas, representative of the Italian agriculture, are characterized by different climatic conditions, crops types, water availability and source, irrigation strategies and distribution rules. These differences will

help to demonstrate the robustness of the implemented methodology. Both areas have experienced drought periods in the past, especially the Capitanata one, where due to very high temperatures and water shortages during August 2012, the production of tomatoes has been affected with a reduction in yield that exceeds 20-25%. Similarly, during the drought of July 2017, the account of the damages has exceeded the billions of euro (Coldiretti, https://www.coldiretti.it/).

The Chiese Irrigation Consortium (www.consorziodibonificachiese.it) extends for an area of about 20 000 ha in the Pianura

Padana Plain near the Lake Idro in the North of Italy. The area is intensively cultivated with summer crops as maize and forage and winter wheat. During summer, crops are sustained by high irrigation volumes provided on a priori fixed turn every 7 and a half days from April to September. Irrigation is provided with the surface technique withdrawn the water from a dense channel network. An average irrigation volume of about 1200 mm is provided during the crop season, over a mean precipitation value of 250 mm.

In contrast, the Capitanata Irrigation Consortium (www.consorzio.fg.it) extends for an area of 50 000 ha in the Puglia region in the South of Italy. The area is characterised by intensive agriculture with summer wheat and tomatoes, and autumn and winter fresh vegetables. Irrigation is provided on demand through a pressurized network pumping water from two reservoirs,



**Table 1.** Spatial dataset sources and characteristics

| Variable | Dataset | Retrieval technology | Revisit time | Product grid | Source |
|---|---|---|---|---|---|
| Precipitation | ECMWF ERA5 | Reanalysis model | hourly | 31 km | Hersbach et al. (2014) |
| | Ground stations | Rain gauges | hourly | | |
| Soil moisture | SMOS | Passive (L band) | 1-2 days | 25 km | Kerr et al. (2010) Al Bitar et al. (2017) |
| | SMAP | Passive (L band) | 2.2 days | 36 km | Entekhabi et al. (2014) |
| | ESA-CCI | Active, Passive, Combined | 1.2 days | 0.25° | Gruber et al. (2019) |
| | Copernicus (Sentinel1) | Active (C band) | 4.1 days | 1 km | Bauer-M. et al. (2018) |
| | FEST-EWB hydrol. model | Energy-water balance | hourly | 30 m | Corbari et al. (2011) |
| LST | Landsat 7-8 | Thermal infrared | 8-15 days | 30 m | Skokovic et al. (2018) |
| | MODIS | Thermal infrared | daily | 1 km | Sobrino (2001) |
| Vegetation index (NDVI, NDWI, SAVI, EVI) | Landsat 7-8 | VIS NIR | 16 days | 30 m | Skokovic et al. (2018) Corbari et al. (2020) |
| | MODIS | VIS NIR | 500 m | 8 days | Myneni et al. |

from April to October. The fields are averagely watered with a mean seasonal amount of 600 mm plus a rainfall mean of about 150 mm, mainly with drip and micro-sprinklers techniques.

### 2.2.1 Data sources

The considered dataset encompasses different variables (precipitation, soil moisture, land surface temperature and vegetation indices) which are obtained from different sources (remote sensing, ground data and hydrological land surface model) at different spatial and temporal resolutions (Table 1). In particular, a low spatial resolution time series and a high one from 2000 to 2020 are considered, to be able to produce a robust statistic based on 20 years of data. Therefore, the anomalies are

computed for the different variables and for the different data sources, and compared. This is done in order to understand if high resolution data are able to capture agricultural drought at large scale, and in particular if high resolution information is needed for agricultural drought monitoring and irrigation aqueduct management, even with lower temporal scales.



### 2.2.2 Precipitation

Precipitation data are provided by ground stations and from reanalysis data (Table 1). In particular, ground data are obtained
from the stations managed by the private association Meteonetwork, plus the stations of the official network of regional ARPA
Puglia for the Capitanata Consortium and ARPA Lombardia for the Chiese Consortium (Fig.2). Data are available at hourly
scale from 2013 to 2018 for South of Italy while from 2005 to 2018 for North of Italy. The data are interpolated through the
inverse distance weighting function (Corbari et al., 2009).

Precipitation data are also extracted from the ERA5 database provided by ECMWF (European Centre for Medium-Range
Weather Forecasts, Hersbach et al., 2020). ERA5 is a reanalysis dataset which provides the description of land water and energy
cycles over several decades from 1950. ECMWF reanalysis combines the high-resolution ECMWF land surface model driven
by the downscaled meteorological forcing from the ERA5 climate reanalysis (Hersbach et al., 2020) with observations from
across the world into a globally complete and consistent dataset. A main advantage of ERA5 is the horizontal resolution, which
is enhanced globally to 31 km compared to 80 km ERA-Interim, whereas the temporal resolution is hourly as in ERA5. ERA5
is freely available through the Copernicus Climate Change Service (Hersbach et al., 2014) from 1950 to present. Beck et al.
(2017) showed that this dataset has the best performances among several other reanalysis datasets and satellite data.

### 2.2.3 Satellite Soil Moisture datasets

The Soil Moisture Ocean Salinity (SMOS) is a European Space Agency (ESA) Earth explorer mission launched in 2009 (Kerr
et al., 2010). It focuses on land Surface Soil Moisture (SSM) and ocean salinity, employing full-polarization L-band (1.4 GHz)
passive microwave observations. For this part of the electromagnetic spectrum, surface soil moisture related emissions are
stronger than for higher frequencies (as the commonly employed C-band). The coarse resolution SSM data (25 km) are avail-
able twice every day, through an ascending and a descending orbit (06:00 and 18:00 local time, respectively). The MIRCLF31
Level3 product v4 used for this study (Al Bitar et al., 2017) was downloaded from the *Centre Aval de Traitement des Données
SMOS* (CATDS) processing center. The data were filtered for Radio Frequency Interference (RFI) probability ($<0.9$) and $\chi^2$
index probability ($<0.9$) (Al Bitar et al., 2012).

Launched in 2014, the Soil Moisture Active Passive (SMAP) mission from the National Aeronautics and Space Adminis-
tration (NASA) featured both (active) radar and (passive) radiometer, with an L-band technology, only the radiometer data are
operational. The SMAP mission provides surface soil moisture products at 36 km (Entekhabi et al., 2014).

The ESA Climate Change Initiative (CCI, Gruber et al., 2019) dataset is the final product of a wide effort aimed at stan-
dardizing and merging a large range of different SSM datasets gathered throughout the years into one global database. On a
first level, two main Active- and Passive-technology datasets were obtained, by aggregating a number of comparable products:
AMI-WS, ASCAT-A and ASCAT-B for the ESA-CCI Active product and SSMR, SSM/I, TMI, AMSR-E, Windsat, SMOS
and AMSR2 for the ESA-CCI Passive product. In this process, data from different datasets were merged by setting a common
reference for each category (ASCAT for Active and AMSR-E for Passive) towards which each dataset maxima and minima had





to be adjusted (Liu et al., 2012). Finally, the ESA-CCI Combined dataset is obtained by scaling both databases to a common product (GLDAS Noah, Dorigo et al., 2017).

    The Copernicus Surface Soil Moisture 1 km Version 1 product (SSM1km) is the result of the Sentinel-1 SAR backscatter observation in band C. Being and Active-type observation, it is provided in saturation percentage, and at the relatively high 1 km spatial resolution. Revisit times are slightly longer than those of the coarser-resolution products, around 4 days (Bauer-
Marschallinger et al., 2018).

### 2.2.4   Satellite Land Surface Temperature

Satellite Land Surface Temperature (LST), as the results of the energy balance of the fluxes between the atmosphere, soil and vegetation, has been retrieved from MODIS low resolution data and from the high-resolution LANDSAT series.

    MODIS satellite data (http://ladsweb.nascom.nasa.gov/index.html) on board the operative satellites Terra and Aqua have
been analyzed from 2000 to 2020. In particular, LST products from the MODIS/Terra LST/E Daily L3 Global 1-km Grid product (MOD11A1) with a spatial resolution of 1 km and a daily temporal resolution are used. The product is based on a split window algorithm to obtain LST data from the combination of MODIS 31 and 32 bands. The average LST error is 1 K in the range from -10 to 50 °C (Wan et al., 2004). LST data have been filtered based on the quality control tags, thus considering only the pixels that have a QC values of 00 and 01, mainly overcoming the cloud issues (Wan, 2008).

High resolution Landsat 7 ETM+ and Landsat 8 satellite data have also been considered (USGS, http://earthexplorer.usgs.gov/). Landsat 7 Enhanced Thematic Mapper Plus (ETM+) bands are acquired at 60-meter resolution and then resampled to 30 meter in the delivered data product. One thermal band 6 is also present. Landsat 8 Operational Land Imager (OLI) and Thermal Infrared Sensor (TIRS) images consist of nine spectral bands with a spatial resolution of 30 meters for Bands 1 to 7 and 9. Thermal bands 10 and 11 are collected at 100 meters but resampled to 30 meter in delivered data product. The Landsat 8
satellite images the entire Earth every 16 days in an 8-day offset from Landsat 7. Land surface temperature is computed using the Single Channel surface temperature retrieval algorithm, based on the Radiative Transfer Equation which can be written in the thermal infrared region as (Jiménez-Donaire et al., 2020; Jimenez-Munoz et al., 2014). The atmospheric correction is performed considering the NASA website (NASA, http://atmcorr.gsfc.nasa.gov/) (Barsi et al., 2005). The surface emissivity is computed as a function of the vegetation fraction, according to (Jiménez-Donaire et al., 2020).

### 2.2.5   Satellite Vegetation Indices

Different vegetation indices (VIs) might be retrieved from remote sensing data to describe the evolution of an agricultural drought, taking advantage of their characteristics of detecting vegetation changes.

    The Normalized Difference Vegetation Index (NDVI) is the most widely used, representing an indication of plants greenness and their photosynthetic activity, responding to changes in the chlorophyll content and the intracellular spaces in spongy
mesophyll of plant leaves. NDVI is computed as the normalized difference between the near infrared (NIR) and visible red





reflectance:

$$NDVI = \frac{NIR - Red}{NIR + Red} \qquad (4)$$

Additionally, to NDVI, Soil Adjusted Vegetation Index (SAVI) was also considered as an index able to reduce the soil background effect (Huete, 1988). It is computed as:

$$SAVI = \frac{b_{850} - b_{660}}{b_{850} + b_{660} + L}(1 + L) \qquad (5)$$

where the sub-index of b refers to bands wavelength, in nanometers, and *L* accounts for first-order soil background variations and in our case is computed as the average vegetation cover of the image which vary from 1 (image fully covered by vegetation pixels) to 0 (image fully covered by bare soil pixels).

Normalized Difference Water Index (NDWI Gao, 1996) is a satellite-derived index from the Near-Infrared (NIR) and Short
Wave Infrared (SWIR) channels, allows estimating the vegetation water content based on shortwave infrared band reflectance increases or decreases. It is sensitive to variations in the water content (absorption of SWIR radiation) and spongy mesophyll (reflectance of NIR radiation). NDWI is computed as:

$$NDWI = \frac{NIR - SMIR}{NIR + SWIR} \qquad (6)$$

Soil background effects can be important in case of partial vegetation cover (Gao, 1996). Drought and water stress are not
the only factors that can cause a decrease of NDWI values/anomalies. Change in land covers or pests and diseases can also be responsible for such variation of the signal, as well as senescence.

Finally, the enhanced vegetation index (EVI) is also considered, in order to detect the minimization of the canopy-soil variations and its sensitivity over dense vegetation.

The Moderate Resolution Imaging Spectroradiometer (MODIS) data used in this study (Table 1) are the 8-day composite
(the best quality daily reflectance data of the 8-day period), 500-meter dataset MOD13A1. The products of NDVI, NDWI, EVI and SAVI are directly used. At high spatial resolution, the vegetation indices are computed from Landsat 7 and 8 satellite images at 30 m spatial resolution.

### 2.2.6 Crop yield data

Crops yield estimates are available from the Italian National Statistics Institute (ISTAT, https://www.istat.it/it/agricoltura?dati).
Regional aggregated data of total yield from the main cultivated crops in the area and cultivated hectares. For the Capitanata area, the province of Foggia in Puglia Region is selected, covering almost the 95% of the Consortium area; while the Province of Brescia covers almost all of the Chiese Consortium Area, but which extends further for one quarter of its area.


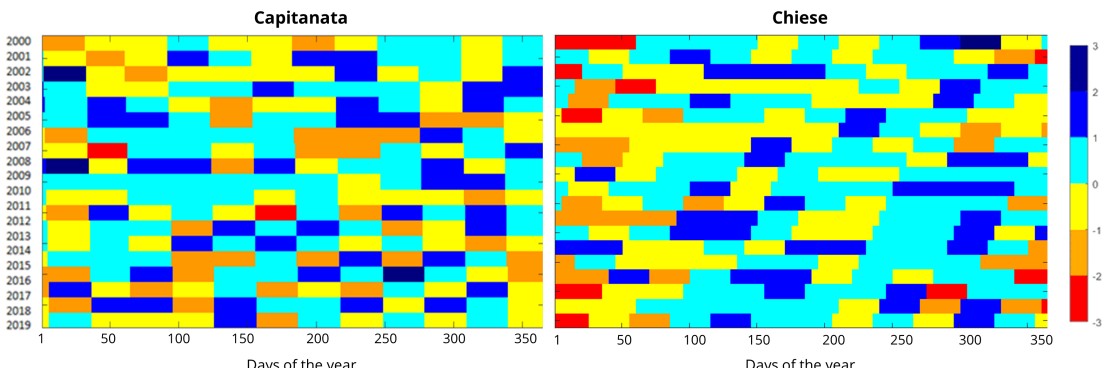

**Figure 3.** SPI-1 monthly index for Capitanata and Chiese Consortia

## 3 Anomalies analysis

### 3.1 Precipitation SPI-1

Precipitation SPI-1 is calculated of both case studies considering as input data the measurements from regional ground stations network (Fig.2) and from ERA5 reanalysis dataset. A good agreement is obtained between the series with a RMSE of 0.33 mm and Pearson coefficient of 0.92 for the Capitanata Consortium, and similarly for the Chiese Consortium area with a RMSE of 0.52 mm and a $r$ of 0.97. In Figure 3, the SPI-1 is reported for both case studies for the ERA5 dataset. From a first look in the period of interest for summer crops cultivation (e.g. and irrigation), no clear deficit situation can be identified in the spring-

summer seasons as well as extreme rainfall droughts periods. The Capitanata area is in genereal more subject to rainfall spring drought conditions than the Chiese area. Looking at the figure we also see that there have been important winter droughts in the last years.

It is important to note that the SPI-1 index is the result of a statistical process where precipitation is fitted with a gamma distribution on non-zero values per month of the year and then normalized. For dry climates, where zero precipitation values are

common, such as the case of South Italy, the calculation of SPI might be skewed (Wu et al., 2007). Therefore, the application of SPI is critical and its interpretation must be done properly. On the contrary, the use of a Precipitation Anomaly index (PA) might lead to an overestimation of drought conditions, due to the detection of negative anomalies that simply represent the anomaly for all the non raining day (zero precipitation value) but this negative anomaly does not represent always an effective precipitation deficit.

### 3.2 Soil Moisture anomalies

Soil moisture anomalies are calculated using the available satellite data (Table 1) for each available date based on monthly variations at daily scale, as these are more in line with the temporal horizon of irrigation decision making for farmers and Irrigation Consortia.



ESA-CCI datasets are the most complete series from 2000 to 2019 for both case studies, instead SMOS and SMAP start
from 2010 and 2015 respectively. In Figure 4 the different SMA values for all datasets are reported for the Capitanata area. A
significant variability among the results is immediately detectable, with a stronger signal in the ESA-CCI Combined and ESA-
CCI Active datasets with negative SMAI in summer and positive SMAI values in winter. SMAI from ESA-CCI Combined
is also characterized by some severe drought conditions. This last dataset is also the longest one, allowing for a more robust
anomalies analysis. Both datasets share the aggregation structure of the ESA-CCI datasets, with the homogenization of different
products into a unique dataset. The resulting, clearly evident SMAI trends, as opposed to the much less obvious trends for other
(non-aggregated and non-homogenized) datasets, seem to suggest that only enlarging the reference data pool it is possible to
identify these trends.

Indeed, SMOS and SMAP anomalies do not show a seasonal trend as clear as that of the ESA-CCI datasets. No further
contribution seems to be provided by higher-resolution information, as even the anomalies in SM Copernicus Sentinel 1 data,
at the high spatial resolution of 1 km (against the >25 km for SMOS and SMAP) do not quite capture any significant trend.
Low similarity is also visible among the two root zone datasets of FEST-EWB and SMOS root zone. The two datasets differ
mainly in the spatial resolution, from the 30 m of the FEST-EWB to the 25 km of SMOS root zone, and also in the soil depth
representativeness, from 60 cm of FEST-EWB to 1-2 m of SMOS (Al Bitar et al., 2013).

Even data density seems to have no major bearing on the matter, as both the ESA-CCI datasets and the root-zone ones
share an almost-daily data frequency but show different identifications of the seasonal trends. On the other hand, lower-time-
frequency datasets (SMOS, SMAP and Sentinel 1) equally display little trace of seasonal trends. Also, the length of the different
datasets does not seem to impact on the results, being the SMOS one available from 2000 while Sentinel 1 and SMAP only
from 2014 and 2015, respectively.

The same comparison is performed for the Chiese Consortium and similar results are obtained in terms of data products,
with consistent heterogeneity across the different datasets. However, some differences emerge, with indications from ESA-
CCI Combined and Active less clear than for the Capitanata case study. The other ESA-CCI dataset (the Passive) highlights
consistent water abundance in the early 2010s, similarly only in part to SMOS rootzone SMAI data, whereas SMOS and SMAP
seem to indicate small drought presence in that period. On top of these mixed interactions between the datasets, seasonal trends
are (also in this case) not easy to detect. The extended analysis is reported in Appendix A.

A quantitative comparison among the different soil moisture products is also carried out, calculating the Pearson correlation
coefficient among the different SM datasets anomalies, for both case studies. This is computed taking each product at a time
as reference. Considering the differences in spatial resolution (25 km for ESA-CCI, 40/50 km for SMAP and SMOS, 1 km
for Sentinel 1 to 30 m of FEST-EWB), in satellite reading frequency and retrieval algorithm, generally low r values are found
with a better correlation in the Chiese area (Table 2). One main distinction between the two test cases is the wetness level:
the Chiese case study sees higher average yearly rainfalls (>750 mm/yr, with more than 200 rainy days a year) with respect to
Capitanata (550 mm/yr, with 150 rainy days a year). This seems to reflect positively on the correlation between the different
SMAIs, probably because it corresponds with a more stable, less peaked SM trend, easier to reproduce from products working
at different resolutions and with different algorithms. In contrast, the high heterogeneity of the Capitanata Consortium (e.g.

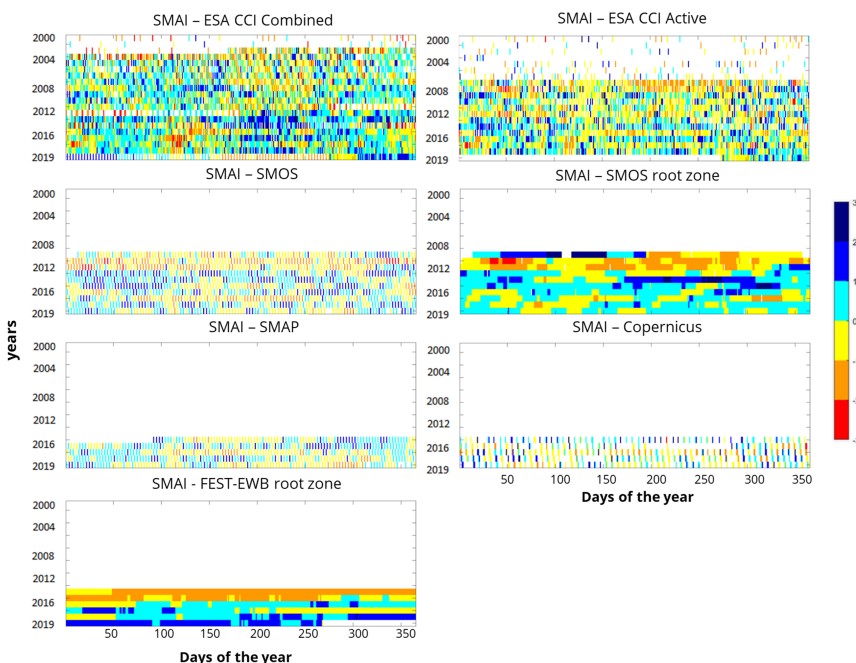

**Figure 4.** SM anomalies in the Capitanata area with data from: ESA-CCI Combined, ESA-CCI Active, SMOS, SMOS root zone, SMAP, Copernicus, FEST-EWB

high contrast between hot dry bare soil pixels and vegetated wet pixels) leads to lower average correlations among products
anomalies.

    The results show how the ESA-CCI datasets are the ones best correlated to each other overall, even if this result is mainly driven by the fact the ESA-CCI Combined dataset includes the other two (Active and Passive). Similarly, the data from SMOS, SMAP and S1 are weakly correlated to ESA-CCI data and among each other. The data of the FEST-EWB model are similarly correlated to the others, with a uniform distribution over the two case studies.

To further understand the discrepancies among the SMAI anomalies computed from the different products, monthly correlations are also analyzed, by taking as reference the most complete dataset, namely the ESA-CCI Combined. In Figure 5, the Pearson correlations are reported for both case studies. In general, for the Chiese area, little difference is observable among the products in the different months of the year, probably because of the higher water amounts (due both to higher rainfall depths and more frequent and abundant irrigations) which smooth out seasonal variations. Differences are visible only for SMOS-
rootzone, with differently related anomalies and the only instance of negative correlation (June) against ESA-CCI Combined data.

    For the Capitanata Consortium, a greater variability of correlations is obtained between different months and different products. Especially the high spatial resolution products show very low correlation in summer with the low-resolution reference dataset ESA-CCI Combined, with numerous instances recorded of negative correlations. This might be related to the inability





**Table 2.** Pearson correlations among SMAI from different datasets for Capitanata (upper half table) and Chiese (lower half table)

| Capitanata<br>Chiese | ESA-CCI,<br>Combined | ESA-CCI,<br>Active | ESA-CCI,<br>Passive | SMOS | SMOS,<br>root-zone | SMAP | Copernicus | FEST-EWB |
|---|---|---|---|---|---|---|---|---|
| ESA-CCI, Combined | n/a | 0.65 | – | 0.37 | 0.17 | 0.34 | 0.31 | 0.2 |
| ESA-CCI, Active | 0.79 | n/a | – | 0.41 | 0.16 | 0.37 | 0.41 | 0.22 |
| ESA-CCI, Passive | 0.78 | 0.38 | n/a | – | – | – | – | – |
| SMOS | 0.53 | 0.51 | 0.43 | n/a | 0.37 | 0.41 | 0.52 | 0.16 |
| SMOS, root-zone | 0.26 | 0.26 | 0.12 | 0.51 | n/a | 0.26 | 0.22 | 0.3 |
| SMAP | 0.46 | 0.42 | 0.48 | 0.84 | 0.32 | n/a | 0.47 | 0.3 |
| Copernicus | 0.51 | 0.55 | 0.39 | 0.64 | 0.23 | 0.55 | n/a | 0.15 |
| FEST-EWB | 0.47 | 0.53 | 0.38 | 0.65 | 0.31 | 0.47 | 0.34 | n/a |

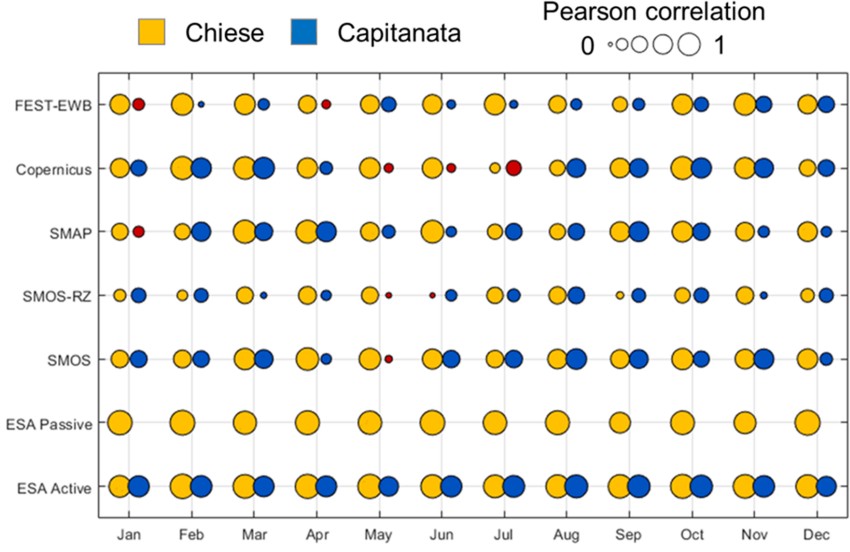

**Figure 5.** Monthly Pearson correlations between the different SMAI products and the ESA-CCI Combined SMAI data, for Chiese and Capitanata. Bigger circles correspond to higher correlations values, while red circles are indicative of negative values.

of the low-resolution product to distinguish between the few irrigated areas and the dry areas, as the higher resolution product can. This is also noticeable in the SMOS Rootzone correlations.





## 3.3 Land Surface Temperature anomalies

LST series retrieved from MODIS at 1 km of spatial resolution are averaged over the analyzed area and used to calculate LST
anomaly. In Figure 6 the anomalies are reported for the Capitanata area. Positive values of LSTA represent warmer days, where
the LST is higher than the mean and water stress is liable to occur, whereas negative values identify the colder-than-average
days. A higher number of highly-negative/negative anomalies are detectable during the summer season. This might be due to
the fact that only the 40% of the area is cultivated with highly irrigated crops (mainly tomatoes) while the remaining part is
cultivated with rainfed wheat or bare soil (Corbari and Mancini, 2022).

It is worth to remember that a monthly anomaly is computed, so that the trend seasonality is removed with no default low
values in autumn/winter and high values in spring/summer.

LST monthly anomaly is also calculated using the average at basin scale of Landsat (L7 and L8) images (Fig.6). A good ad-
equacy is observable between the two datasets, with a RMSE equal to 0.1 and a Pearson coefficient of 0.73; while significantly
more data are available for MODIS series.

Similar results are obtained for the Chiese area with no specific seasonal stress periods. However contrary to Capitanata, the
summer period is usually characterized by several days of high positive anomalies. This due to the fact that the Chiese area is
mainly cultivated with irrigated corn. A good adequacy is observable between the Landsat and MODIS dataset, with a RMSE
equal to 0.1 and a Pearson coefficient of 0.73; while significantly more data are available for MODIS series.

These results agree with the validation of LST estimates (i.e., not anomalies) as obtained by Jimenez-Munoz et al. (2014)
and Corbari et al. (2020) who compared LANDSAT 7 and 8 against ground data with mean absolute differences close to zero
with an RMSE of 3.0 K and of 2.0 K for LANDSAT 7 and 8, respectively. As well, Duan et al. (2019) obtained for MODIS
daytime data RMSE values around 2 K.

## 3.4 Vegetation indices anomalies

Vegetation indices anomalies are finally considered to detect and monitor the impacts on vegetation growth and productivity of
environmental stress factors, especially plant water stress due to drought. It is still an open problem in literature which index
best represents the crops stress and dynamics (Huete et al., 1997). Hence, anomalies are calculated for all the vegetation indices
NDVI, NDWI, EVI and SAVI from MODIS and compared.

In Figure 7, the results are shown for the Capitanata Consortium. An almost identical trend is obtained for NDVI and SAVI
with negative and negative anomalies which are detected in the same periods. In particular, negative values are more frequent
during summer and early autumn season, even though in the last years high positive values are registered. The results are
consistent with the typical agricultural practices of the Capitanata area where during summer most of the area 40% is not
cultivated. Similar trends are also observable between NDVI and EVI, even though fewer negative anomalies are observable,
especially during the early autumn. Instead, when comparing NDVI and NDWI anomalies, some peaks with opposite sign
are observed (Fig.7). As it is well known, NDVI is strongly correlated to leaf area color and low values are synonymous of
vegetation stress with a reduced photosynthetic capacity, while NDWI is linked with the crops leaves' moisture. The opposite


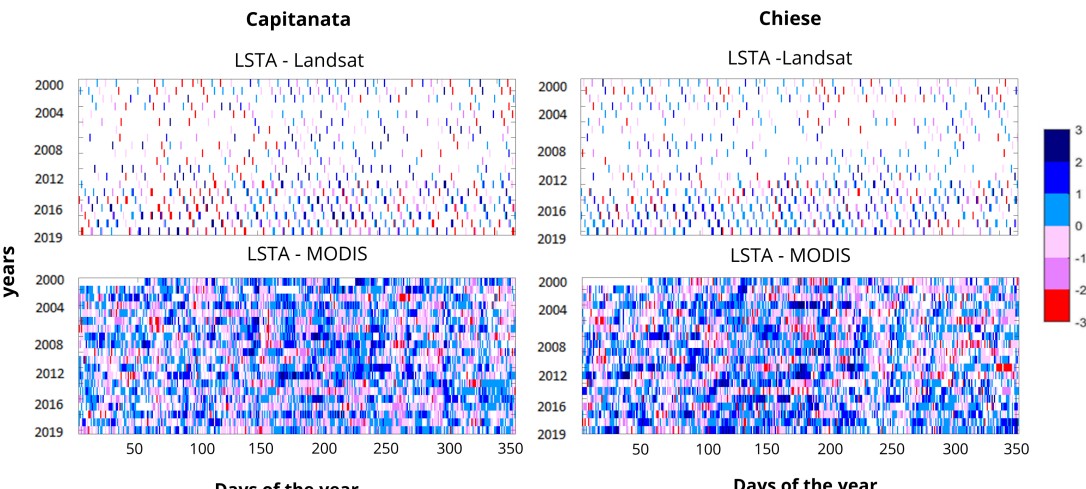

**Figure 6.** LSTA anomalies from MODIS and Landsat for Capitanata and Chiese

peaks are caused by precipitation events that produce an immediate increase in the NDWI values but not in the NDVI, that is always affected by a delay in the response. However, Gu et al. (2008) found that both indices might be suitable for vegetation drought analysis.

Moreover, while NDVI estimates might be influenced by soil background conditions and non-drought stress conditions, such as plant disease (Xue and Su, 2017), similar anomalies changes are detected with SAVI index, which on the contrary is not 365    influenced by soil effects (changes in soil color or soil moisture). The EVI index which accounts for the interaction of the atmospheric conditions (Liu and Huete, 1995), is similarly correlated to NDVI and SAVI anomalies.

The comparison among the different vegetation indices for the Chiese area, while showing the same discrepancies as for the Capitanata area, is reported in Appendix A.

These results are confirmed from the Pearson correlation coefficients, computed taking each product a time as reference 370    (Table 3). As expected, NDVI anomaly is perfectly correlated with the SAVI one, which in turn shows a good correlation with EVI anomaly (0.65). Extremely low correlations are confirmed among NDWI and the other indices anomalies, with values less than 0.5.

NDVI has then been used for the computation of the ADMOS index. It should be noted that vegetation indices could catch not only vegetation anomalies due to drought conditions but also the senescence phase. However, this is reflected only in the 375    NDVIA while the ADMOS index will not be impacted because the vegetation drought condition is considered only in the last step on the index calculation after that all the other three anomalies are considered.

The spatial resolution effect is further analyzed by comparing the NDVI anomalies from MODIS (at 500 m) with the ones computed from Landsat (L7 and L8) images (at 30 m) (Fig.8). Similar results are obtained for both case studies with a Pearson coefficient of 0.72 and a RMSE of 0.99.


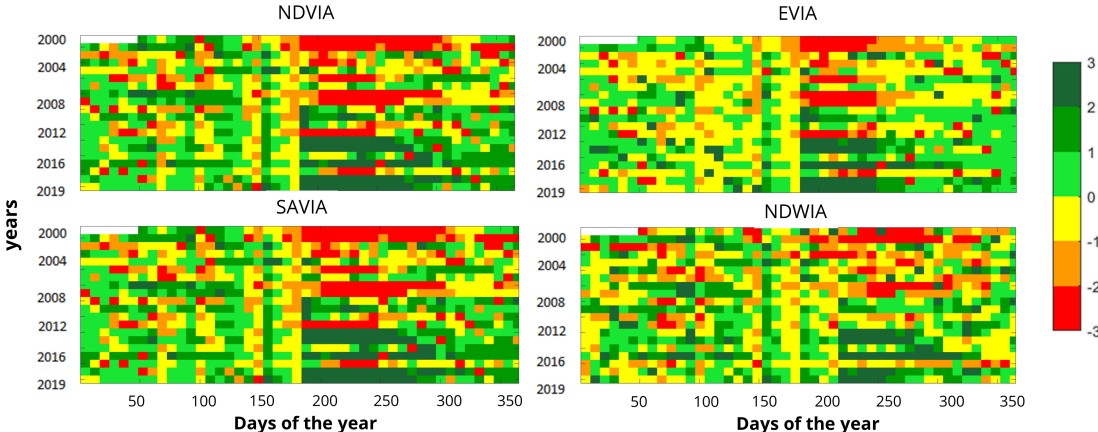

**Figure 7.** Vegetation indices anomalies (NDVI, EVI, SAVI and NDWI) for the Capitanata Consortium

**Table 3.** Pearson correlations among different vegetation indices anomalies for Capitanata (upper half table) and Chiese (lower half table)

| Capitanata / Chiese | NDVI | EVI | SAVI | NDWI |
|---|---|---|---|---|
| NDVI | n/a | 0.65 | 1.00 | 0.49 |
| EVI | 0.5 | n/a | 0.65 | 0.35 |
| SAVI | 1.00 | 0.5 | n/a | 0.49 |
| NDWI | 0.44 | 0.44 | 0.44 | n/a |

## 3.5 Synchronicity and correlations among anomalies

The synchronicity or asynchronicity among the different variables' anomalies are further analyzed during the different years to understand how a precipitation drought could lead to a moisture one, which might or not be transferred to the temperature and vegetation droughts. The temporal consistency between a pair of variables is defined building a contingency table: a positive agreement between the two anomalies (both positive or both negative) or a disagreement when an anomaly is positive and the other negative, and vice versa. Then, the percentage of days of the four possible combinations of the contingency tables are computed. The five possible combinations of variables' anomalies are shown for each month in Figure 9 for Capitanata area and in Figure 10 for the Chiese one.

Differences between the case studies are immediately visible in the temporal consistencies, especially when SM anomalies are compared with precipitation, NDVI and LST. This confirms the complex behavior of droughts with widespread asynchronicities also in spatial patterns.





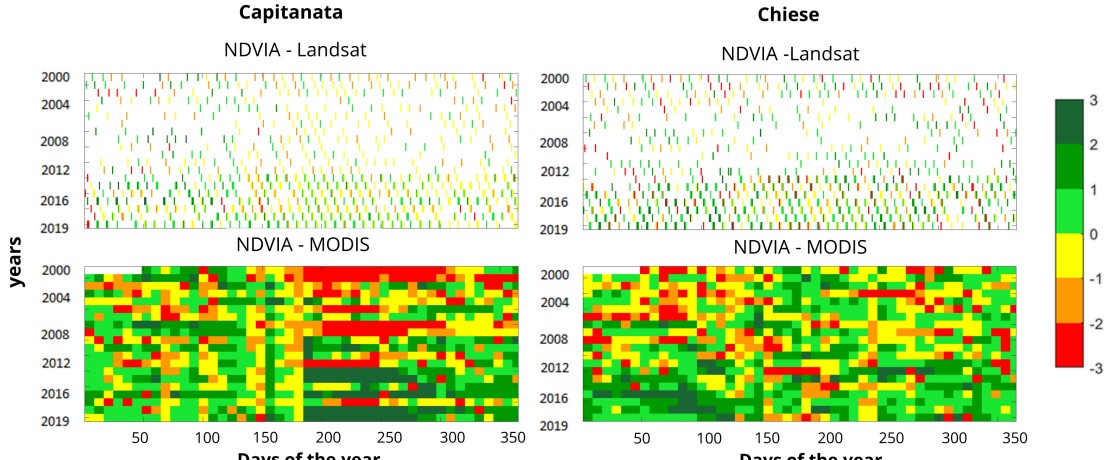

**Figure 8.** NDVI anomalies from MODIS and Landsat for Capitanata and Chiese

In the Capitanata Consortium, considering SPIA and SMA, a consistency of 40% is found with both negative anomalies during the summer months, and of 50% for only a negative SM anomaly for the same period. This is in line with the characteristics of this agricultural area which is generally very dry in summer with few precipitation events and only 45% of the area is cultivated with irrigated crops. During the other months of the year, the conditions of no drought or of opposite signs of the anomalies with a negative SPIA prevail. When precipitation deficit occurs, e.g. negative values of SPI are observed, the fields

are irrigated so SM increases and SMA takes positive values. Hence, the Pearson correlation coefficient over the whole period is between SPI and SMA is quite low (0.2) with positive values. SMA and NDVIA agree for 40%, during summer, indicating that the area is partially characterized by an NDVI and SM deficit. Consistency of SMA negative values is equal to 50% that reaches the 60% in July and August, showing that the SM drought is not reflected into a vegetation stress. This situation might

be due to the fact that the vegetated areas are mainly irrigated and the NDVI index obtained from the MODIS dataset at 1 km of spatial resolution is able to catch these differences, while the soil moisture from the low spatial resolution dataset (25 km) is not detecting the differences between irrigated and not irrigated fields, but only the average area value. Moreover, it might be related to the different time response of the index to drought (e.g. precipitation is the first while vegetation is the last). Lastly, it should be noted that if an energy limited condition occurs, vegetation is less affected by negative anomalies on soil

moisture (on the contrary to what happens during water limited conditions). During the other months of the year, a situation of positive anomalies (SMAI and NDVIA) is prevailing and some rare vegetation stress events are present with no soil moisture drought conditions. This particular situation might be related to energy, heat stress, or pests and diseases with no limitation on water content availability. Being the Capitanata Consortium located in the South of Italy, energy limitations are probably not occurring, while heat stress can surely be a limiting factor. In general, a low and positive Pearson coefficient (0.23) is found

between SMA and NDVIA all over the years.





A high seasonal dynamic of anomalies consistencies is also observable between SMA and LSTA, with a consistency of 50% indicating both drought conditions as well as only SM stress during the summer months. This is in line with the obtained results between SMA and NDVIA, as well as similar explanations could be given on the response of LST to SM conditions. A negative Pearson correlation is found (–0.19).

A non-defined trend in the consistencies is instead obtained between SPI and NDVIA as well as between LSTA and NDVIA, with a non-predominant situation along the year. A positive correlation is obtained between SPI and NDVAI (0.18), while a weak negative one between LSTA and NDVIA (–0.05). However, LSTA and NDVIA are negatively correlated only during summer (–0.2), while a positive value is obtained if only winter months are considered (0.33). This might be due to the fact that vegetation stress response delays with respect water deficit and high temperature.

For the Chiese Consortium, a non-defined trend in the consistencies is observable for all variables couples' comparisons, with no predominance of a drought or no drought conditions in any month of the year. The Chiese area is intensively cultivated with forages all the year as well as irrigated, leading to controlled conditions for LST, SM and NDVI.

If SPIA and SMA are compared, the Pearson correlation coefficient over the whole period is positive (0.32) with a lower correlation during summer (0.1). This might be simply explained by the fact that in the Chiese Consortium during the summer 425 season when maize crop is largely cultivated (90% of the area), high amounts of irrigation are used , leading to positive SMA anomalies which, however, might be present with negative SPIA values.

Conversely to the Capitanata Consortium, the Pearson correlation between SMA and NDVIA is slightly negative (–0.07), with SM negative anomalies which are usually not reflected into vegetation stress. This is mainly due to the different time response of vegetation which has a delayed response than soil in respect to water deficit. A clear example of this situation is 430 happening during the year 2004, where a prolonged soil moisture deficit is present with extremely negative anomalies values from March to June, but it is not immediately followed by a vegetation stress deficit which instead happens only from June to September.

Moreover, as for Capitanata area, the accuracy and ability of the current satellite soil moisture products to detect the changes in precipitation or irrigation has been shown (Paciolla et al., 2020) to be quite weak. Outside of the irrigation period, most 435 satellite datasets show physical consistency with (either natural or artificial) water inputs only around 50% of the time, with few exceptions (75% for the ESA-CCI Combined dataset). During the irrigation season, these performances improve (the inconsistency instances decrease by a factor higher than 2), with a minor contribution of the irrigation inputs (which explain 20% of the SSM dataset variability). Thus, anomalies in SM retrieved from satellite are only partially respondent of rainfall or irrigation effects.

As no prevailing trend in consistency, all the other variables correlations are very low, being the negative the ones between SPI and LSTA (–0.01), SPI and NDVIA (–0.04) as well as SMAI and LSTA (–0.01), while a slight positive correlation is obtained between LSTA and NDVIA (0.1). This might be due to the fact that vegetation stress response delays with respect to water deficit and high temperatures.



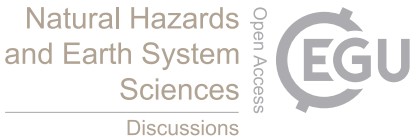

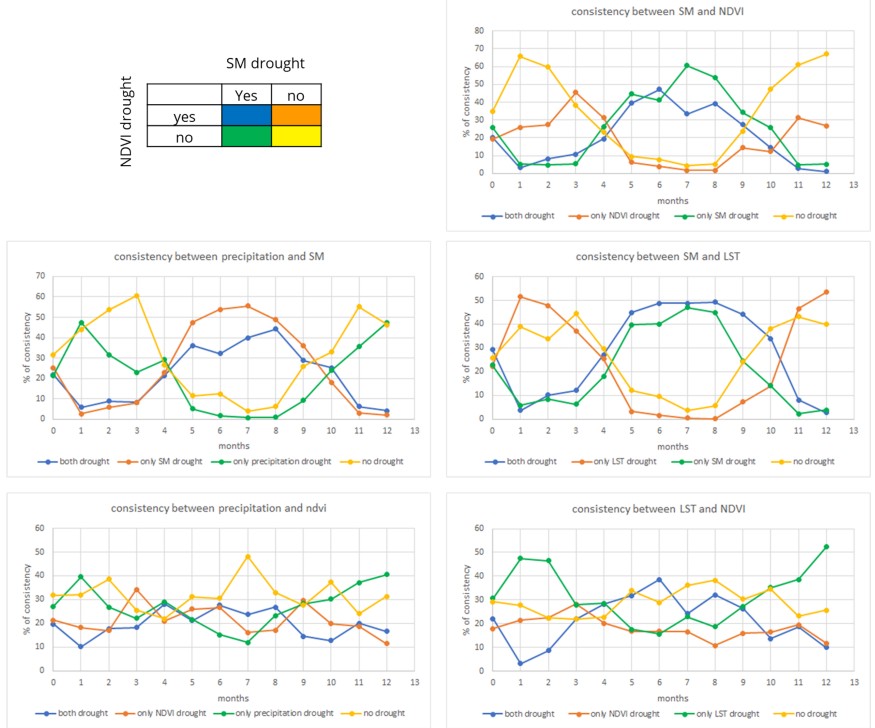

**Figure 9.** Synchronicity among the different variables' anomalies in the Capitanata Consortium

## 4 Agricultural Drought monitoring system

The ADMOS is finally evaluated at daily time scale comparing different combinations of anomalies products: the low spatial resolution datasets combinations (SPI, with SM from ESA-CCI Combined or Active, SMOS, SMAP or SMOS root zone, LST and NDVI from MODIS) and the two high resolution combinations (SPI, SM from Copernicus and LST and NDVI from Landsat or MODIS). This allows studying the reliability and the response of different input data on the drought index values. To easily compare how the level of dryness changes in time, the cumulative drought index curve is evaluated for each year. In

Figure 11, the comparison is reported for both case studies in terms of daily and yearly cumulated values. All the curves have a common trend: from the begin of the year till March the curves have gentle slopes, then from March to October they are very steep due to drought conditions, and at the end of the year they return flat. This behavior agrees with the crop seasonality and with the irrigation period. In general, a good accordance is visible among the different series, being the high-resolution series able to accurately reproduce the low-resolution behavior, even if for a shorter time period of analysis. Comparing the different

EO products combinations, no relevant differences in the combined drought index cumulated values are observed in both case studies. However, some differences are visible, in particular, when the SM product of SMOS root zone is used during the years 2011 and 2012 in Capitanata and 2010, 2011, 2012 and 2017 in the Chiese. Another year of discrepancies is the 2015 in the Capitanata area when all SM products produce a lower cumulated value than the two ESA-CCI products.





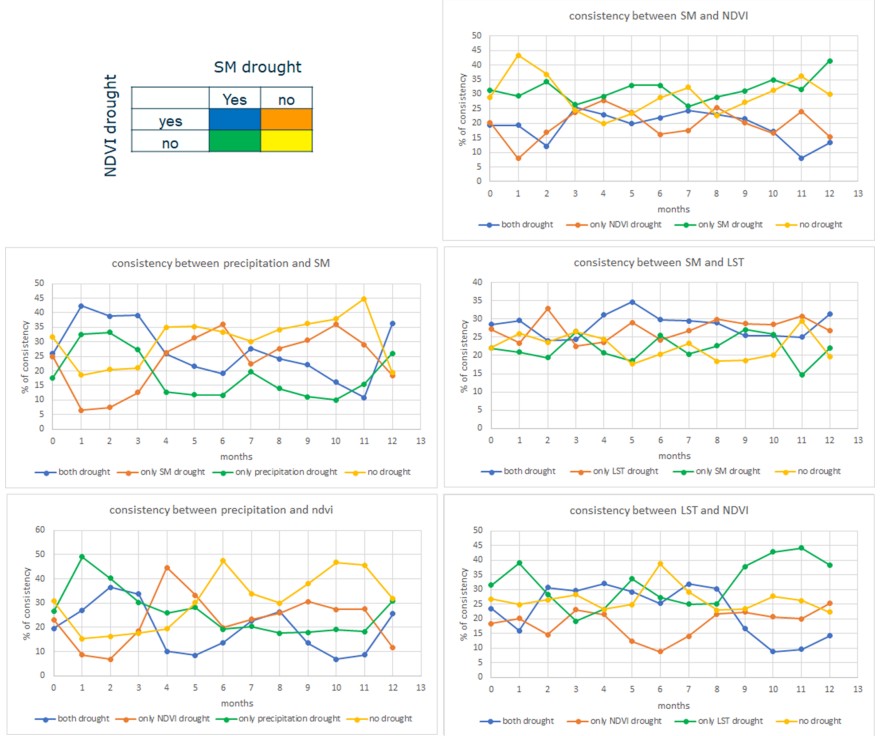

**Figure 10.** Synchronicity among the different variables' anomalies in the Chiese Consortium

In Capitanata area, the driest years are identified as 2003, 2011, 2012 and 2017, while the wettest years are 2010 and 2014
(Fig.11). A first confirmation of the capability of the combined drought monitoring system to detect agricultural drought con-
ditions is the comparison with local analyses by the Italian farmers association Coldiretti (https://www.coldiretti.it/), which
reports very high temperatures and water shortages during August 2012, July 2017 and summer 2003 and reduction in toma-
toes productions that exceeds 20-25%. Furthermore, a comparison of drought indications is performed against the European
Drought Observatory (EDO, https://edo.jrc.ec.europa.eu), as being the operative reference at European Commission level. An
agreement is found between the two drought indicators in identifying the driest year for 2012 and 2017, while the EDO is not
reporting a vegetation drought in 2012 and no soil moisture drought in 2003. This last year however was one of the driest years
in Italy and Europe (Diodato and Bellocchi, 2008; Musolino et al., 2018).

Similarly for the Chiese Consortium, 2010 and 2014 are identified as very wet years, while very dry years are 2003, 2007,
2017 and 2019. ADMOS is in general agreement with EDO estimates, except for the year 2017 where no vegetation drought
is detected by EDO and for the year 2010 where EDO is no reporting soil moisture positive anomalies. It should be noted that
the EDO system is using the FPAR index for vegetation instead of NDVI, and the soil moisture estimates come from a land
surface model (Sepulcre-Canto et al., 2012).


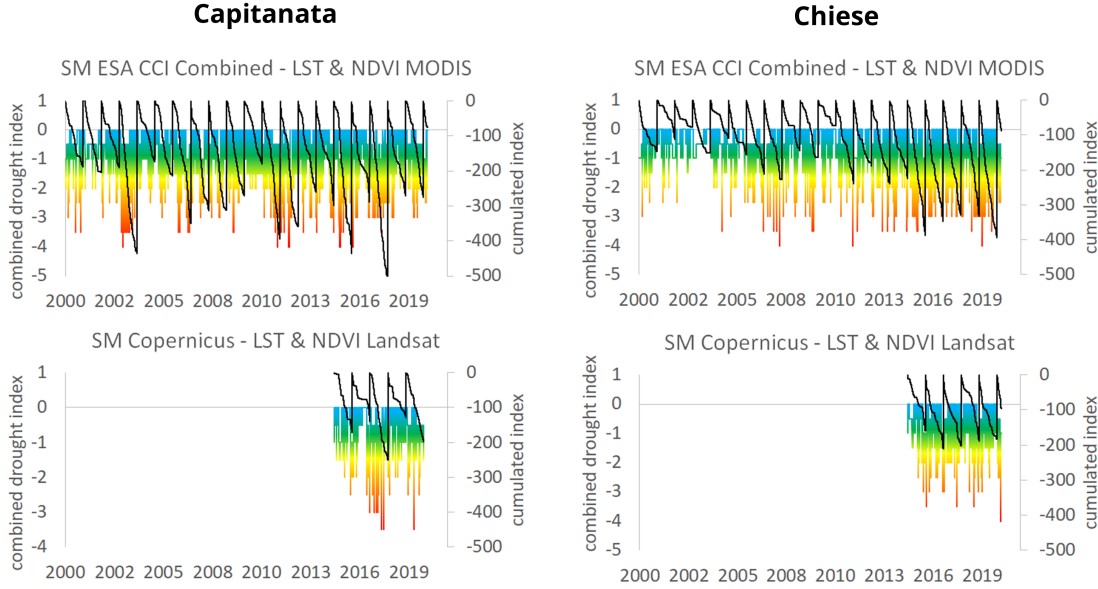

**Figure 11.** ADMOS daily values and cumulative yearly values for Capitanata and Chiese Consortia for the high and low resolutions series

## 4.1 ADMOS relationship with irrigation management and crop yield

Irrigation volumes are a way to provide a reliable assessment of the drought index values, as one can expect a higher volume
of water used for irrigation during stressed conditions.

In the Capitanata Consortium, observed daily cumulated irrigation volumes are provided by the aqueduct from 2006 to 2016, except for 2008 and 2009, from the 1st of April to 31st December for each year. These values are compared to the daily cumulated ADMOS in the same periods of irrigation volumes availability. A difference of about 20 000 000 m$^3$ of irrigation water is detectable between the most irrigated year, i.e. 2012 and the least irrigated one, i.e. 2009. By comparing the behaviour
of ADMOS and irrigation volumes, it is immediately visible the high consistency among the years, with the driest years (2012 and 2011), characterized by the lowest ADMOS values, being the years with the highest water volumes used for irrigation (Figure not shown). This is confirmed from the scatterplot of the cumulative drought index values and irrigation volume at the end of each year (Fig.12). A negative correlation is shown, where an increase in the drought index means an increase of dryness conditions, meaning that more irrigation is required for both the low spatial resolution dataset (SPI, SM from ESA CCI
Combined, LST and NDVI from MODIS) and the high resolution one (SPI, SM from Copernicus and LST and NDVI from Landsat).

Furthermore, the ADMOS is also compared to the cumulated volumes of the total water supplied to the fields, by summing the precipitation volume to the irrigation one. A slightly positive correlation is obtained, meaning that during all years, the total amount of water (irrigation + precipitation) provided to crops is similar. This might be explained by the fact that in the





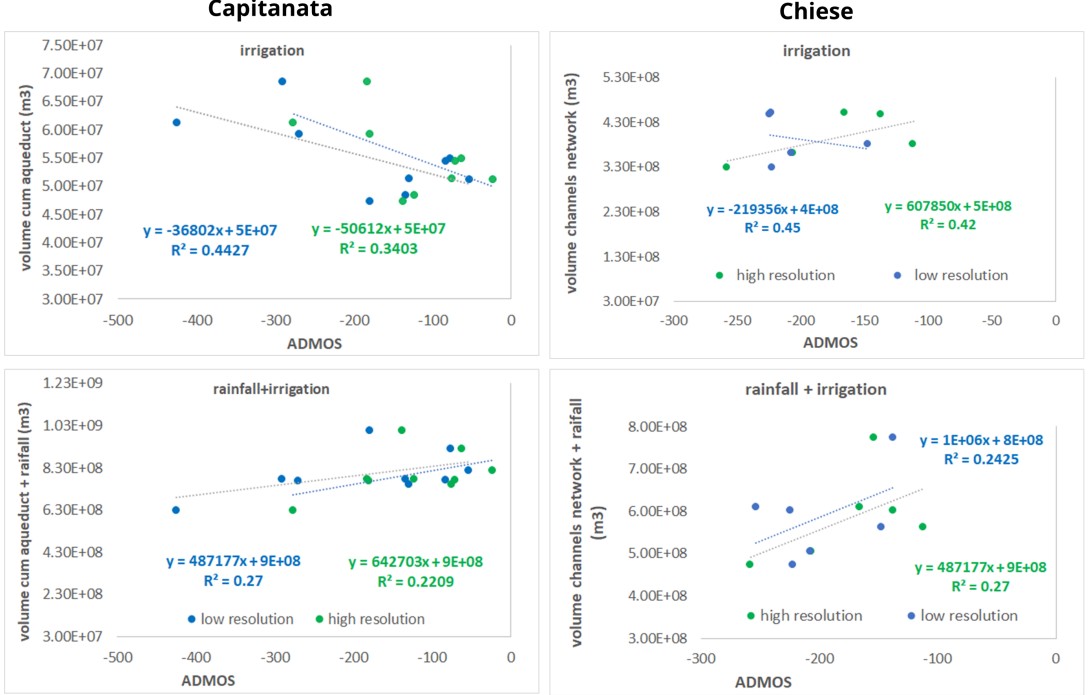

**Figure 12.** ADMOS correlation with irrigation volumes and with rainfall plus irrigation for the Capitanata and the Chiese Consortia

Capitanata Irrigation Consortium the farmers pay a quota for m$^3$ of water used per ha; so that during wetter years, lower irrigation volumes are provided.

An opposite situation is observed in the Chiese Irrigation Consortium (Fig.12) where the cumulated ADMOS and the irrigation volumes provided by the channels network at the end of each year are compared in a scatterplot. Almost zero correlation is found, with an almost constant amount of irrigation volumes which is provided to the crops every year independently from the

drought or not condition. Consequently, if rainfall volumes are added to the irrigation ones, a positive correlation is obtained. This is immediately explainable by the water costs in the Chiese Consortium, where the farmers pay a fixed quota every year per ha independently from the volumes of water used. Hence, during the rainy years, too much water is probably provided to the crops. This is true for both high and low resolutions datasets.

As a further validation of the implemented index as well as to understand the impact that possible drought conditions might

have on crops production, a crop specific correlation is analysed in the two case studies by comparing the Consortium crop yields and the ADMOS values (Fig.13).

In the Capitanata Consortium, the crop yield seems to be almost uncorrelated with the ADMOS index. This might be due to the fact that the area is highly irrigated during summer and that the amount of irrigation is capable of contrasting the drought conditions allowing to maintain an almost constant production over the years. The oscillations in contrast might

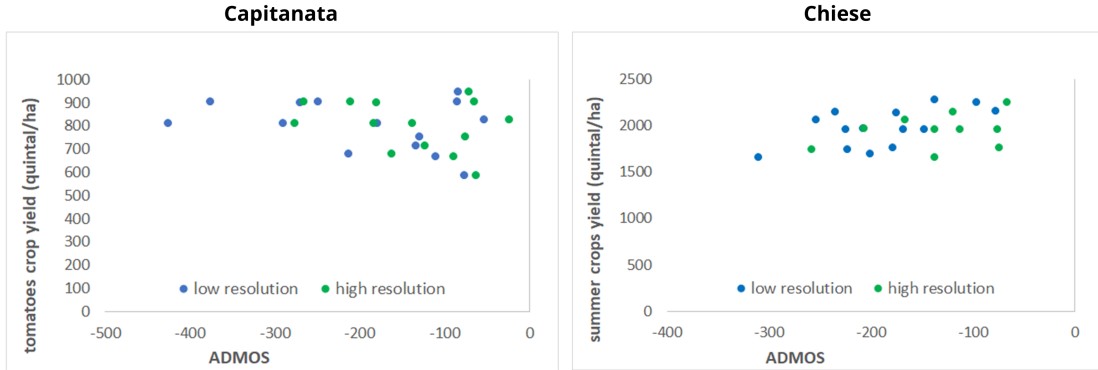

**Figure 13.** ADMOS and crop production correlation, for tomatoes in the Capitanata Consortium and for summer crops in the Chiese Consortium

be related to other factors, as changes in pest and diseases. A common problem in the tomato production in the area is the *Phytophthora infestans*, which is especially affecting the crops during conditions of water stagnation. Even if the agricultural drought monitoring system reaches high negative values, these are persistent for few days, which are not enough for affecting the crop production. Moreover, opposite climate extremes as frost conditions or hailstorms during the spring, might also strongly affect crop production, but are not detected by the implemented drought index.

In the Chiese Consortium, a similar null correlation is obtained between the summer irrigated crops yield and the ADMOS index. As well, this area is highly irrigated with an excess of water used, which is not affecting the production. This might be instead affected by other external factors.

The correlation is thus having a spread preventing from the conclusion of a direct link with crop production in the analysed case studies.

## 5 Discussion and conclusions

Following the principles of CAP to improve irrigation management to reduce crop damages and to enhance economic income, an EO-based drought system, ADMOS, has been developed for the operative management of irrigation networks, demonstrating its capability in providing reliable estimates of drought conditions on a daily basis, correctly identifying the wet and dry years. This ADMOS might help irrigation districts managers and farmers to activate the preventive protection actions to try to avoid water volume and crop yield losses. The methodology has been applied in two cases studies: the Chiese and Capitanata Irrigation Consortia in Italy, over 20 years of analysis for the period from 2000 to 2019.

The robustness of the developed drought system has been evaluated by assessing the effect of drought events on the effectively used irrigation volumes and yearly crop yields at Irrigation Consortium Scale, in Southern and Northern Italy. This quantitative verification of the methodology, especially against irrigation volumes, represents an innovative point in respect to





the generally implemented drought monitoring systems, mostly not validated with external data but relying only on comparisons among indices (Wood et al., 2015; Sheffield et al., 2012; van Dijk et al., 2013) or economic impacts evaluation (Ding et al., 2011). In the Capitanata Consortium, a negative correlation is obtained between the yearly cumulative drought index values and irrigation volumes, where an increase in the drought index means an increase of dryness condition, meaning that more irrigation is required; conversely in the Chiese Consortium a zero correlation is obtained with an almost constant amount

of irrigation volumes provided to the crops every year independently from the drought condition. This might be explainable by the irrigation water payment plan, where in the Capitanata Irrigation Consortium the farmers pay a quota for $m^3$ of water used per ha, so that during wetter years, less irrigation volumes are provided; while in the Chiese Consortium, the farmers pay a fixed quota every year per ha independently from the volumes of water used, so during the rainy years, too much water is probably provided to the crops. In both analyzed areas, crop yields seem to be almost uncorrelated to the drought index, probably due to

the fact that the area is highly irrigated during summer and that the amount of irrigation is capable of contrasting the drought conditions allowing to maintain an almost constant production over the years.

One of the strengths of the developed procedure relies on the capability of capturing the temporal evolution of the agricultural drought process, by firstly considering the meteorological drought and then the agricultural one, vegetation water stress and crop drying. This methodology improves the traditional analysis, which are generally analysed by considering only soil

moisture anomalies (van Hateren et al., 1990; Sadri et al., 2018). In fact, the results of this work clearly show that asynchronies may exist especially between soil moisture anomalies and vegetation or land surface temperature anomalies, with negative SMAI and positive NDVIA. This situation might be especially relevant when conditions of energy limited to vegetation growth are present, so that vegetation is less affected by soil moisture changes. This is due to the multifaced processes of vegetation, which is not driven only by water availability (Nemani et al., 2003).

These results are in general agreement with the findings of Wanders et al. (2017), Bachmair et al. (2018) and Tijdeman et al. (2022), who highlighted that, generally, droughts cannot be described by one single indicator but there is the need first to select the correct physical index for detecting a drought type and secondly to use different drought indices to identify specific conditions to avoid disagreements among information sources (Parsons et al., 2019; Sheffield et al., 2012).

Another strength of the developed methodology is that the ADMOS is fully based on remote sensing data, considering

different products of the same variable which differ on the sensing techniques, the spatial and temporal resolutions. The combinations of different products lead to different absolute drought values but these are almost all consistent (e.g. positive or negative anomalies).

In particular, seven to eight soil moisture products anomalies have been compared and generally low Pearson correlation values are found with a better correlation in the Chiese area, probably due to higher average yearly rainfalls which correspond

with a more stable, less peaked SM trend, easier to reproduce from products working at different resolutions and with different algorithms. The results show how the ESA-CCI datasets are the ones best correlated to each other overall, even if this result is mainly driven by the fact the ESA-CCI Combined dataset includes the other two (Active and Passive). The reason why satellite SM does not show encouraging results may lie, among other reasons, in the fact that satellite soil moisture products are limited to the top few cm of soil, whereas a soil moisture drought assessment is ideally based on observations over the entire root zone,





as well as the presence of coastlines in nearby the Capitanata area or the water bodies in the Chiese Consortium. The physical inconsistencies that satellite SSM data shows against water input (both rainfall and irrigation) make them not fully suitable to be used alone in drought detection.

On the other hand, high consistency has been obtained among different vegetation indices, with NDVI anomalies showing a high Pearson correlation especially with EVI and SAVI anomalies. This result in terms of anomalies confirms the general

behaviour of this multitude of vegetation indices which, even if they produce slightly different results in terms of vegetation water status, does not significantly change the drought detection (Shahabfar et al., 2012).

Another point worth of notice is the insertion of the LST anomaly into the drought monitoring system to detect the plat water stress in advance in respect to plant drying. This relies on the fact that the LST is the driving factor of the partitioning between latent and sensible heat fluxes in the energy balance. This LST consideration is an improvement of the generally used

agricultural drought indices.

This paper tried to respond also the issues related to spatial resolution of satellite data answering to the question if high resolution information is needed for drought monitoring and irrigation aqueduct management. The obtained results indicate the advantage of using high-resolution data in respect to low resolution data, as being able to catch the same behavior in terms of agricultural drought detection at Irrigation district scale, and potentially offers important new advantages for applications

even down to field scales. This might be particularly relevant in complex landscapes, where low resolution data might produce contrasting signals.

*Data availability.*   Data freely available online.

**Appendix A**

Soil Moisture anomalies have been computed also for the Chiese case study, with the graphic results provided in Figure A1. As

for the Capitanata case study, consistent heterogeneity can be found across the different datasets, with indications from ESA CCI Combined and Active even less clear than in the other case study. Earlier (2004-2008) SMAI estimations from ESA CCI Passive seem to highlight consistent water abundance, although the droughts registered in the 2010-2019 period are perceived from other datasets in a mixed way: while SMOS rootzone shares some critical values around year 2010, SMOS and SMAP data seem to indicate small drought presence in that period. Generally, seasonal trends seem to be visible only in some years of

the SMOS rootzone dataset, which shows some points in common with the modelled FEST-EWB rootzone SMAI: non-drought conditions at the onset and ending of summer, with more severe conditions in between, although present only in some years. The high spatial resolution of SM Copernicus Sentinel 1 data seems to provide small further information. One general trend is the relatively high Pearson correlation coefficient among the different SM datasets anomalies, as opposed to the lower values registered for the Capitanata case study. The higher wetness of the Chiese case study may act as an equalizer, bringing closing

the gaps between the different retrieval technologies behind each product.

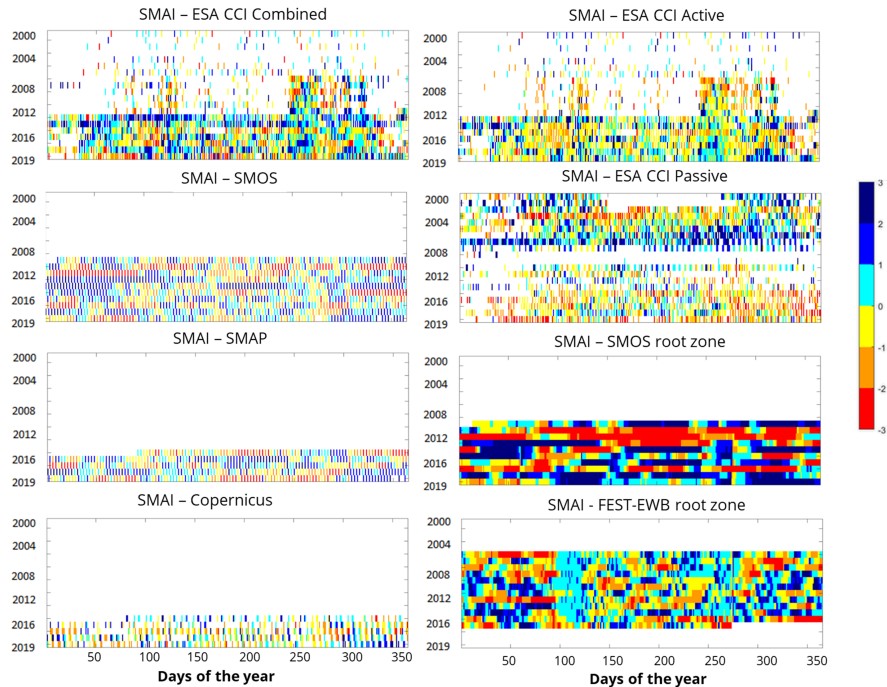

**Figure A1.** SM anomalies in the Chiese area with data from: ESA-CCI Combined, ESA-CCI Active, ESA-CCI Passive, SMOS, SMOS root zone, SMAP, Copernicus, FEST-EWB

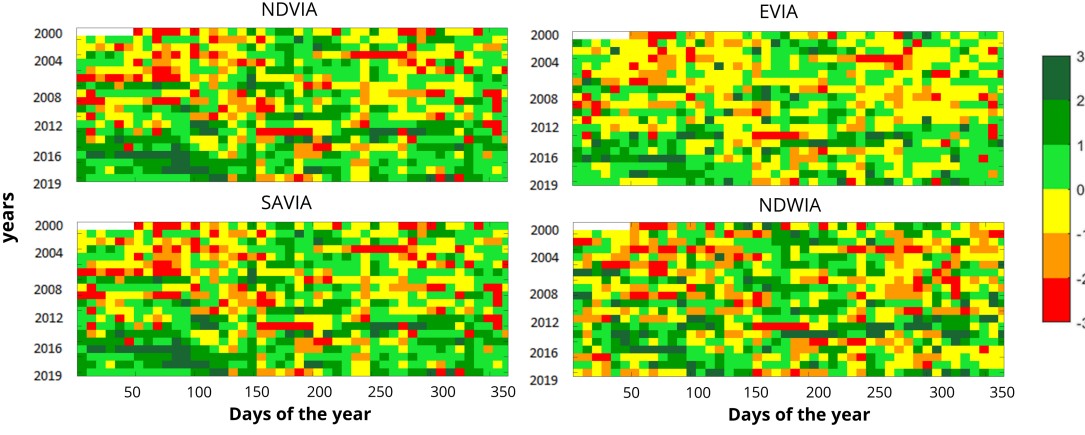

**Figure A2.** Vegetation indices anomalies (NDVI, EVI, SAVI and NDWI) for the Chiese Consortium

*Author contributions.* Conceptualization: C.C. and A.A.; Data elaboration: C.C. and G.R.; Methodology: C.C., N.P. and A.A.; writing and editing: C.C. and N.P.; validation: N.P., C.C. and A.A. All authors have read and agreed to this version of the manuscript.



*Competing interests.* The authors declare no conflict of interest.

*Acknowledgements.* This work has been developed under the projects: RET-SIF real time soil moisture forecast for smart irrigation (EU
ERANETMED Programme) – 2018-2021 funded by the Italian Ministry of Education (MIUR). Hersbach et al. (2014) was downloaded
from the Copernicus Climate Change Service (C3S) Climate Data Store. The results contain modified Copernicus Climate Change Service
information.

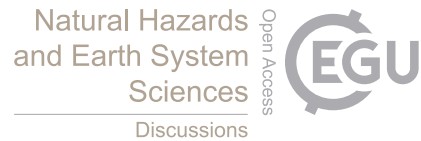

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
