# Peer review of "Multi-scale EO-based agricultural drought monitoring system for operative irrigation networks management"

_Natural Hazards and Earth System Sciences, 2022_

## Author Comment (AC1)

**CC1 –**

Authors' aim is to establish a daily system, but the time scales of your input variables are monthly,16 days, 8-days and the like. I think it has great uncertainties that you resample monthly and 16-days scales into daily scale. Here, my suggest is that only use precipitation as the input, there are many daily precipitation data, and you can avoid the monthly scale. Also, the spatial resolutions of data authors used are very different, I think you can clarify how to resolve this problem, because your small study area.

Line 110, this paragraph, I think author should modify it into a flow chart, because you aim is to establish a drought monitoring system. It will be more clear for potential readers. Actually, soil moisture is an efficient indicator for vegetation drought, but in this manuscript "2) soil moisture shortage is evaluated by the soil moisture anomaly (SMA), 3) vegetation drying is identified with a land surface temperature anomaly (LSTA)", I think it is unreasonable. And, my suggestion is use soil moisture and root-zone soil moisture as the vegetation water stress, LST anomalies as the heat stress and NDVI anomalies as the vegetation conditions.

Answer: we thank the reviewer for this question, which helps us to clarify this point that has been also raised by other reviewers. The anomalies are computed using the daily data (when available according to satellite acquisition time) and normalized according to the long-term mean daily values. The only index computed at monthly scale is the SPI-1 which, by its own definition, is related to 1 month anomaly.

Regarding the temporal scales of operation of the irrigation systems, these are different from North to South Italy, mainly due to the irrigation management schedule (weekly and up to daily, respectively). In the Chiese area, a priori fixed-schedule is defined before the start of the irrigation season with turns every 7.5 days from April to September. On the other hand, in the Capitanata area, irrigation is provided on demand so that irrigation volumes are regulated on a daily scale variation. The ADMOS index is updated daily, so that the time scale evolution is consistent with water management.

Regarding the methodology section, this will be rephrased to better explain all the steps in the analysis, also with the aid of a flow chart. The ADMOS indicator is computed assigning specific values reported in Fig.A, according to increasing drought conditions. The procedure is divided into four subsequent steps: 1) precipitation deficit, 2) soil moisture anomaly, 3) land surface temperature anomaly and 4) vegetation index anomaly. We believe the use of different data sources of different origin to be a major strength of this work. In fact, many studies on drought analyze the indicators using a single source of data, but as noted, the uncertainty of satellite information can be significant. Therefore, the possibility for the indicators to have an ample data pool, drawing from multiple data sources, strengthens the methodology developed in this work. Finally, we removed the "surplus of water" conditions (when SPI-1 was positive) in order to avoid confusion, as it did not took part in the calculation of the cumulated ADMOS.

[Figure]

Fig.A Improved version of Fig.1 from the manuscript, detailing the ADMOS workflow/flowchart

---

## Author Comment (AC2)

With interest I have read this manuscript addressing a new drought index. The topic issued by the manuscript "Multi-scale EO-based agricultural drought monitoring system for operative irrigation networks management" is relevant and the structure of the manuscript is well organized. I personally think the manuscript has potential to be published, but there some aspects that need to be clarified by the authors, before I can recommend this work for publication. Based on this  is that I recommend major revisions.

General comments:

- Based on the title of this manuscript, please further discuss the added information of the ADMOS index for both (Chiese and Capitanata) operative irrigation networks. In particular, how this index can improve the water management? Is it applicable only for irrigated crop regions? On this later aspect it is also important to be clear in regard to the objective of ADMOS, it is meant only for monitoring or also for predicting droughts?

Answer: ADMOS is conceived as a monitoring tool, to be employed in order to follow more closely the evolution of possible drought dynamics. The index might be useful in a on-demand irrigation scheme (as the Capitanata case study), to know the evolution of water availability as well as crops conditions and to infer the possible increase or decrease of irrigation water requests from the farmers allowing to better manage the water at Consortium scale.

ADMOS data requirements, encompassing easy-to-obtain EO observations, make its applications extremely simple to all kinds of crop regions. The use over irrigated areas which was described in this study is mainly due to the possibility of involving the irrigation volumes in the monitoring evaluation, but does not restrict the validity of ADMOS to such cases thanks to the world-wide availability of remote sensing data. The ADMOS index, as described in this work, is meant for monitoring of droughts even in real-time with daily to weekly scale. We will include this in the new introduction.

- Another aspect which needs to be clarified and justified is related with the performing part of the evaluation of the ADMOS index against crop yield, is this evaluation valid if crops receive water through irrigation?

Answer: the evaluation over irrigated crops was a precise intent of this work. More than the link between ADMOS values and final yield, we wanted to show how the different irrigation regulations, management and volume of water used for irrigation in the two consortia correlate (or not) with the annual drought conditions: high correlation in areas where water is provided on demand at relative higher cost (Capitanata irrigation consortium) and lower correlation where water is provided on a fixed-schedule and on a fixed yearly cost, independently of the meteorological conditions (Chiese irrigation consortium).

- From the manuscript, it is not clear on what temporal scale the ADMOS index works, is the monthly, weekly? In addition and related to this, discuss what temporal scales which are needed by operative these different irrigation networks.

Answer: we thank the reviewer for this question, which helps us to clarify this point that has been also raised by other reviewers. The anomalies are computed using the daily data (when available according to satellite acquisition time) and normalized according to the long-term mean daily values. The only index computed at monthly scale is the SPI-1 which, by its own definition, is related to 1 month anomaly.

Regarding the temporal scales of operation of the irrigation systems, these are different from North to South Italy, mainly due to the irrigation management schedule (weekly and up to daily, respectively). In the Chiese

area, a priori fixed-schedule is defined before the start of the irrigation season with turns every 7.5 days from April to September. On the other hand, in the Capitanata area, irrigation is provided on demand so that irrigation volumes are regulated on a daily scale variation. The ADMOS index is updated daily, so that the time scale evolution is consistent with water management.

- A particular concern is related with the high vs. low resolution analysis. The results are based on a small sample of data which show a considerable dispersion in the relationship between ADMOS and rainfall and rainfall+irrigation. Why the authors seek for a linear relationship? Should the relationship be linear? How robust or significative are these results? Please discuss the potential drawbacks of all these considerations in the analysis.

Answer: The main idea is to compare the anomalies and ADMOS estimates which come from the available remote sensing datasets, which have a opposite performance in terms of temporal and spatial resolutions: high spatial resolutions mainly correspond in low temporal one (especially true for LST) and vice-versa. Moreover, very few applications of drought indices are based on high spatial resolution (30 m) which can provide very detailed information. We will add this comment in the discussion section, highlighting that it's a first application which could be improved by enlarging the time series of data.

Specific comments:

- Improve in general the figure caption descriptions.

Answer: We will improve the figure captions enlarging the description of Figure content.

Figure 3. SPI-1 monthly index for Capitanata and Chiese Consortia using the ERA-5 dataset from 2000 to 2020

Figure 4. Daily SM anomalies in the Capitanata area with data from: ESA-CCI Combined, ESA-CCI Active, SMOS, SMOS root zone, SMAP, Copernicus, FEST-EWB from 2000 to 2020

Figure 5. Monthly Pearson correlations between the different SMAI products (ESA-CCI Active, SMOS, SMOS root zone, SMAP, Copernicus, FEST-EWB) and the ESA-CCI Combined SMAI data, for Chiese and Capitanata from 2000 to 2020. Bigger circles correspond to higher correlations values, while red circles are indicative of negative values.

Figure 6. Daily LSTA anomalies from MODIS and Landsat for Capitanata and Chiese from 2000 to 2020

Figure 7. Daily vegetation indices anomalies (NDVI, EVI, SAVI and NDWI) from MODIS data for the Capitanata Consortium from 2000 to 2020

Figure 8. Daily NDVI anomalies from MODIS and Landsat data for Capitanata and Chiese from 2000 to 2020

Figure 9. Synchronicity among the anomalies of SPI, SMA, LSTA, NDVIA in the Capitanata Consortium from 2000 to 2020

Figure 10. Synchronicity among the anomalies of SPI, SMA, LSTA, NDVIA in the Chiese Consortium from 2000 to 2020

Figure 11. ADMOS daily values and cumulative yearly values for Capitanata and Chiese Consortia: low resolution series (SM ESA CCI Combined – LST and NDVI MODIS) and high spatial resolutions series (SM Copernicus – LST and NDVI Landsat).

Figure 12. Seasonal cumulated ADMOS index correlation with irrigation volumes and with rainfall plus irrigation for the Capitanata and the Chiese Consortia for the low and high resolutions series

Figure 13. Seasonal cumulated ADMOS index and crop production correlation, for tomatoes in the Capitanata Consortium and for summer crops in the Chiese Consortium

- On several parts of the manuscript the word trend is used, but it remains unclear the particular meaning of it. Like for example "seasonal trend" and the examples listed below. Please clarify this aspect across all the manuscript.

Answer: thanks for the comment, which is explained below for each point all the manuscript.

- Why is that the authors start the abstract section with "Drought prediction" if they will focus on monitoring? Of course both topics are of major important for a drought early warning system, but in this case I would suggest using the word monitoring.

Answer: we agree that the monitoring is the main focus of the work, we will change the word in question in the revised version of the manuscript.

- I suggest modify "electromagnetic frequencies" for spectral bands

Answer: thanks, we agree on this modification

- Are they Drought monitoring systems for irrigation regions i other regions of the planet? If they are, I consider that a paragraph related to irrigation networks background and how they use drought indices is needed.

Answer: thank you for this observations, we will add a paragraph on this in the final version of the manuscript. For instance, Luan et al. (2015) designed a drought monitoring/forecasting approach coupled with an irrigation management system, encouraging its application in northern China. Ozelkan et al. (2016) focused more on the sensing aspect of drought detection, discussing the differences between rainfed and irrigated cropland when using satellites to monitor droughts over a semi-arid agricultural area (south-eastern Anatolia, Turkey). Finally, Wu et al. (2015) applied MODIS data over the US corn belt (midwestern US, from Nebraska to Ohio) to identify the best index that could capture the local agricultural drought of 2012. In particular, they identified in the normalized difference infrared index (NDII6) a powerful and promising tool to monitor drought conditions over irrigated land.

1. Luan, Qingzu, et al. "An integrated service system for agricultural drought monitoring and forecasting and irrigation amount forecasting." 2015 23rd International Conference on Geoinformatics. IEEE, 2015.
2. Ozelkan, Emre, Gang Chen, and Burak Berk Ustundag. "Multiscale object-based drought monitoring and comparison in rainfed and irrigated agriculture from Landsat 8 OLI imagery." International Journal of Applied Earth Observation and Geoinformation 44 (2016): 159-170.
3. Wu, Di, John J. Qu, and Xianjun Hao. "Agricultural drought monitoring using MODIS-based drought indices over the USA Corn Belt." International Journal of Remote Sensing 36.21 (2015): 5403-5425.

- On what temporal window is the ADMOS working? Weekly, monthly? Please clarify this

Answer: we thank the reviewer for this question, as also clarified on the previous comment the anomalies are computed using the daily data (when available according to satellite acquisition time) and normalized according to the long-term mean daily values. The only index computed at monthly scale is the SPI-1 as for its definition related to 1 month anomaly.

- Line 76 Is the Global Integrated Drought Monitoring and Prediction System (GIDMaPS, http://drought.eng.uci.edu) still operational?

Answer: the GIMaPS is indeed not operational, with the most recent data referred to 2016. However, we wanted to include it as an effective example of EO-powered monitoring.

- Line 79: CDI index should be an indicator and not an index following the WMO definition as it uses different indices separately and not combined in only one index as the SMADI (Soil Moisture Agricultural Index) for example. It is also important to highlight that the CDI uses different time dates for each variable, which is different to the USDM approach.

Answer: Thanks for the comment, we agree with this definition which will be updated in the revised paper version.

https://www.droughtmanagement.info/literature/GWP_Handbook_of_Drought_Indicators_and_Indices_2016.pdf

- Line 149: An average irrigation volume of about 1200 mm is provided during the crop season, over a mean precipitation value of 250 mm. How are the irrigations estimated?

Answer: they are provided by the irrigation consortia authorities, and are known for the whole areas with daily frequency.

- Line 283: "SMOS and SMAP anomalies do not show a seasonal trend as clear as that of the ESA-CCI datasets." But ESA-CCI considers a longer time period. What is meant with seasonal trend?
- Line 307: "less peaked SM trend" As the authors don´t mention a trend analysis, please clarify what is meant with trend?

Answer: thanks for the comments, which help to correct the improper use of the statistical term "trend". What we mean here is that generally in these two case studies we have negative SM anomalies during summer which can been seen in the ESA-CCI datasets from 2000 to 2010, while it is less visible in the other datasets. We will better explain this in the revised paper version.

- Why the authors use SPI-1 and not SPI-3 or SPI-6?

Answer: if SPI is computed for 1 month it can be used as an indicator for immediate impacts such as reduced soil moisture. For more agriculture related impacts, we are then analyzing directly the effects on vegetation to see eventual crop stress conditions.

- Please specify what is the SMOS Root zone product.

Answer: indeed we have neglected a description of this product in the appropriate section (2.2.1.2), we will add it for the final version. The SMOS root zone soil moisture product is provided by the CESBIO SMOS team (Al Bitar et al. 2013). It employs SMOS L3 surface soil moisture and other data (e.g., NDVI from MODIS, climate data from ECMWF) to derive soil moisture at root level using an exponential filter for the first soil layer and a physical model for the second. This model is a simple formulation of the 1D Richards equations that describe water motion in unsaturated porous media. The product is available with a 0.25° spatial resolution.

- Line 483: Please modify (Figure not shown) for (not shown)

Answer: thanks, we will correct this error

- Figure 12. In this figure is not clear while the ADMOS varies from 0 to -500, please explain clarify with more detail this values as the index form its definition varies from 1 to -4. Add also these details in the figure caption. Is it the accumulation value? Also, use the same amount of decimals for the R2, and I recommend changing the units of m3 to millions of m3 or equivalent.

Answer: Yes, the "ADMOS" shown in fig.12 is the accumulation (or cumulated) value, like the one shown on the right-hand vertical axis of Fig.10. All the other elements you mentioned have been homogeneized.

- Line 532: What are the crop impacts of this "too much water is probably provided to the crops "? Please discus how would the ADMOS help to this.

Answer: in this part, we wanted to highlight how, in some years, the drought conditions were not so important as to require the high water amounts used in reality. Indeed, since ADMOS was not signalling particularly heavy drought conditions, a part of that water could have been saved. It is not within the scopes of this work to estimate the exact amount of irrigation water required in each year, but to show how ADMOS can provide an indirect estimate of the necessity to employ lower or higher irrigation water volumes.

- Line 555: Please clarify this sentence: "In particular, seven to eight soil moisture products anomalies have been compared and generally low Pearson correlation values are found with a better correlation in the Chiese area, probably due to higher average yearly rainfalls which correspond with a more stable, less peaked SM trend, easier to reproduce from products working at different resolutions and with different algorithms." What is meant with 7 to 8 SM products? Is not clear what is meant with less peaked SM trend? What is meant with low Pearson correlation values are found with a better correlation in the Chiese area?

Answer: it refers to the fact that 8 products were involved in the comparisons for the Chiese test case, whereas the Capitanata case did not feature the ESACCI passive product (because of frequent unavailability of the data). By "less peaked SM trend", we meant that the SM values in the Chiese consortium are quite stable at relatively high values, due both to the frequent rainfall and irrigation events. This behaviour is quite homogeneous and probably easier to represent with similar accuracy across different scales than what happens in the Capitanata consortium. Here, both rainfall and irrigation are more sparse in time and more heterogeneous in space, causing the SM time series to have greater variance. Such a complex dynamic could be more difficult to portray, and may be reflected differently in different SM products with different spatial resolutions. We believe this to

be the reason why Pearson correlation values were higher in the Chiese consortium (higher agreement between the different SM products), as opposed to the Capitanata one (higher variability among the products).

- Line 567: Change "plat" for plant

Answer: thanks, we will correct this error

- Figure 11. Please clarify on what temporal scale the index was accumulated? Monthly, Yearly?

Answer: in this case, the ADMOS was accumulated over the whole year, in exactly the same way as was shown in Fig.10.

---

## Author Comment (AC3)

**RC2**

Review of "Multi-scale EO-based agricultural drought monitoring system for operative irrigation networks management"

- This study proposes a methodology to assess drought conditions in two irrigation polygons of Italy based on different data sources obtained from satellite data. My opinion is that this manuscript should not be published because it is affected by several formal and methodological problems. The methodology is not well explained and justified and in general all the manuscript it is very difficult to follow. The use of different data sources of different origin makes difficult to know the connection between meteorological and agricultural droughts. The results are also presented in a very confuse way, with different plots in which it is not possible to obtain a clear message about the relationship between metrics and the evolution of the existing anomalies. Below I am providing specific comments that support my general assessment and my suggestion to reject this manuscript.

Answer: the methodology section will be rephrased to better explain all the steps in the analysis, also with the aid of a flow chart. The ADMOS indicator is computed assigning specific values reported in Fig.A, according to increasing drought conditions. The procedure is divided into four subsequent steps: 1) precipitation deficit, 2) soil moisture anomaly, 3) land surface temperature anomaly and 4) vegetation index anomaly. We believe the use of different data sources of different origin to be a major strength of this work. In fact, many studies on drought analyze the indicators using a single source of data, but as noted, the uncertainty of satellite information can be significant. Therefore, the possibility for the indicators to have an ample data pool, drawing from multiple data sources, strengthens the methodology developed in this work. Finally, we removed the "surplus of water" conditions (when SPI-1 was positive) in order to avoid confusion, as it did not took part in the calculation of the cumulated ADMOS.

[Figure]

Fig.A Improved version of Fig.1 from the manuscript, detailing the ADMOS workflow/flowchart

- 9-14: Very confuse summary of the results. What is a cumulative drought monitoring system? A drought index can be cumulative, but I wonder what are the authors referring in relation to a drought monitoring system. It is not clear what kind of correlation the authors are referring.

Answer: We will improve the abstract in the revised version of the paper. We although believe that all the definitions are present in the paper, where for example it is explained how the time cumulative AMODS index is computed.

Here the new abstract: "Drought prediction is crucial especially where the rainfall regime is irregular and agriculture is mainly based on irrigated crops, such as in Mediterranean countries. In this work, the main objective is to develop an EO-based agricultural drought monitoring index (ADMIN) for the management operative irrigation networks. The ADMIN index considers different levels of drought conditions combining anomalies of rainfall, soil moisture, land surface temperature and vegetation indices. Multiple remote sensing data, which differ on sensing techniques, spatial and temporal resolutions and electromagnetic frequencies, are used and the uncertainty in anomalies computation derived from the use of multiple sources of remote sensing datasets is also discussed. The analyses have been performed over two Irrigation Consortia in Italy (the Chiese and Capitanata ones), which differ for climate, irrigation volumes and techniques, and crop types. The obtained results show an inverse dependency between the cumulated ADMIM and the irrigation volumes in the Capitanata area (which has on demand irrigation), whereas the dependency is much weaker in the Chiese Consortium (where irrigation is provided on a fixed basis, independently from the drought conditions). In both areas, the role of irrigation is critical to sustain production and preserve crop yields, which seem almost uncorrelated to ADMIN.

- 17: There are much better references to refer to drought characteristics and impacts. It seems that the authors have simply cited some papers related to drought... See e.g. IPCC AR6 Chapters 8 and 11 for a summary of drought complexity and implications.

Answer: thanks, but although some of the cited works could be more recent, or up to the point, we selected well-known (and highly-cited) works which describe why droughts are relevant. The selection of references could be a very subjective process. As an example of this, the paper cited first in Line 17, Wood et al. (2005) "summarizes and synthesizes the research carried out under the NOAA Drought Task Force (DTF), including an assessment of successes and remaining challenges in monitoring and prediction capabilities, as well as a perspective of the current understanding of North American droughts and key research gaps. [...] Results from the DTF papers indicate that key successes for drought monitoring include the application of modern land surface hydrological models that can be used for objective drought analysis, including extended retrospective forcing datasets to support hydrologic reanalyses, and the expansion of near-real-time satellite-based monitoring and analyses, particularly those describing vegetation and evapotranspiration". Nonetheless, we agree that some more accurate references could be added throughout the article and will revise this point for the final version of the manuscript.

- 23: In irrigated lands water shortage can be relevant but in rainfed agricultural areas precipitation (but also temperature and atmospheric demand) play a very important role.

Answer: thanks, we will acknowledge this fact in the article.

- 25: Definitively this is not the best to refer to land atmosphere feedbacks and droughts. Again the citations are very poorly selected, which gives a very bad impression as it seems that references are only located randomly in the text to justify the use of references. About this topic, I would recommend

to read Miralles DG, Gentine P, Seneviratne SI, Teuling AJ. 2019. Land–atmospheric feedbacks during droughts and heatwaves: state of the science and current challenges. Annals of the New York Academy of Sciences. Blackwell Publishing Inc., 1436(1): 19–35. ttps://doi.org/10.1111/nyas.13912.

Answer: thanks, we will add this work to the references.

- .25. It should be irrigated agriculture.

Answer: thanks we will add "irrigated" to agriculture

- .28. I would say better: "during the dry season in water limited regions". I would not refer to specific regions.

Answer: thanks we will rephrase it

- .29. One-sentence paragraph? I also find this very disconnected of the context. I suggest to remove this sentence as it does not provide any relevant message.

Answer: thanks, we remodulate this part.

- .35. Again poorly and non-suitable citations. Vicente-Serrano 2006 analyses spatial pattern of meteorological drought but there is nothing on this study on the dynamic of different types of drought. The authors should revisit all the citations of the manuscript. The poor and unsuitable citation approach is a solid formal argument to suggest the rejection of the manuscript.

Answer: thanks, for the observation, we already updated some of the references when answering to the reviewers' comments and will continue to do so for the final version of the manuscript.

- .40. Cite the WMO guidelines for SPI in which it is recommended as a reference drought index.

Answer: thanks, we will add this reference (https://library.wmo.int/doc_num.php?explnum_id=7768)

- .45. The SPEI is perfectly comparable in time and space (as the SPI) Also the Standard Palmer Drought Index is perfectly comparable spatially, so the argument of the authors is not correct. Why is the use of potential evapotranspiration a limitation? I would say that given atmospheric evaporative demand has a relevant influence on drought severity it should be an advantage.

Answer: thanks we agree on this, this line will be corrected in the revised version.

- 48-56: If remote sensing soil moisture is affected by so large uncertainties, what is the justification of its used? The low correlations found among soil moisture datasets presented below even justifies more my assessment.

Answer: Remote sensing products of soil moisture is a state-of-the-art variable, widely diffused in the scientific community and used for an incredibly large number of applications. Historically, SM data was obtained via insitu, point-wise sampling, using either fixed probes or during dedicated field campaigns (Joshi et al., 2016; Dorigo et al., 2011). However precise, this approach is still heavy in terms of required effort and resources (both in terms of time and economic costs) and provides only very locally information. This is an issue, as SM can heavily vary both in time and space because of heterogeneity in the meteorological and bio-geophysical drivers (Famiglietti et al., 2008), affecting the measures representativeness. Furthermore, the measurement density necessary for a meaningful upscaling of SM is itself a matter of discussion (Crow et al., 2012). Instead, remote sensing has eased the monitoring of SM over large areas and at reasonable spatial and temporal resolutions. An extensive review about numerous applications of remote sensing SM (notwithstanding their possible uncertainties) is provided by Babaeian et al. (2019). DIfferent applications show the potentiality of using and comparing SM products from different satellites for drought monitoring. AS Jessica Bhardwaj et al., 2022 who compared SMOS, SMAP and ASCAT, showing that ASCAT is a valuable dataset indicative of agrometeorological drought over Australia.

1. Famiglietti, J. S., Ryu, D. R., Berg, A. A., Rodell, M., & Jackson, T. J. (2008). Field observations of soil moisture variability across scales. *Water Resources Research*, 44, W01423. https://doi.org/10.1029/2006WR005804

2. Crow, W. T., Berg, A. A., Cosh, M. H., Loew, A., Mohanty, B. P., Panciera, R., de Rosnay, P., Ryu, D., & Walker, J. P. (2012). Upscaling sparse ground-based soil moisture observations for the validation of coarse-resolution satellite soil moisture products. *Reviews of Geophysics*, 50, 2011RG000372, L19406. https://doi.org/10.1029/2011RG000372

3. Babaeian, E., Sadeghi, M., Jones, S. B., Montzka, C., Vereecken, H., & Tuller, M. (2019). Ground, proximal, and satellite remote sensing of soil moisture. *Reviews of Geophysics*, 57, 530– 616. https://doi.org/10.1029/2018RG000618

4. Bhardwaj, J.; Kuleshov, Y.; Chua, Z.-W.; Watkins, A.B.; Choy, S.; Sun, Q. Evaluating Satellite Soil Moisture Datasets for Drought Monitoring in Australia and the South-West Pacific. Remote Sens. 2022, 14, 3971. https://doi.org/10.3390/rs14163971

- .57. land surface temperature has been widely used. See e.g. TCI developed by Felix Kogan and the drought monitoring systems (and studies) that use it.

Answer: thanks, we will add this reference to the text to complement the work on LST (Kogan, F. N., 1997: Global Drought Watch from Space. Bull. Amer. Meteor. Soc., 78, 621–636, https://doi.org/10.1175/1520-0477(1997)078<0621:GDWFS>2.0.CO;2.)

1. www.nat-hazards-earth-syst-sci.net/12/3519/2012/ Nat. Hazards Earth Syst. Sci., 12, 3519–3531, 2012 3522 G. Sepulcre-Canto et al.: Development of a Combined Drought Indicator
2. Xiang Zhang, Nengcheng Chen, Jizhen Li, Zhihong Chen, Dev Niyogi, Multi-sensor integrated framework and index for agricultural drought monitoring, Remote Sensing of Environment, 188, 2017, 141-163, https://doi.org/10.1016/j.rse.2016.10.045.

- .71. The optimal solution is really to relate drought objective metrics with impacts and then select the most suited approach. For this purpose, empirical analysis that relates drought indices and impacts is needed.

Answer: we agree with you that such a comparison would be a valuable addition to the workflow of the whole paper. We will add these comparisons to the final version of the work but, in the meanwhile, we attach here some plots, conceptually similar to those in Figure.12 and 13, but featuring the single anomalies (average value in the crops season) instead of the ADMOS, over the Capitanata test case. The strength of ADMOS is particularly

relevant when you compare the single anomalies with the cumulated irrigation volume, where you obtain a positive dependency (at higher SPI values you would expect to use lower irrigation water). Also, with SMA from remote sensing uncertainties in the correlation between SM dynamic and irrigation events. Moreover, comparing the determination coefficients (R2) with those from the figures in the manuscript, the correlation between cumulated volume from the aqueduct and single anomaly is everywhere weaker with respect to the ADMOS. This difference is even wider when considering all water inputs (aqueduct and natural rainfall), with the correlations falling below 0.20. Finally, when looking at the final yield (third figure, to be contrasted with fig.13 from the manuscript), a similar behaviour is observed if considering the ADMOS or single anomalies.

[Figure]

- .83. I wonder if the authors are proposing a drought monitoring system or a drought index. I believe that they are developing a drought index.

Answer: yes, we indeed are proposing a drought index, to be employed for monitoring of possible drought conditions.

- .91. A new drought index should be evaluated with impact data (e.g. crop damages and yields). The volumes of irrigation may be related to several other factors including water availability in reservoir storages, groundwater, etc.

Answer: thanks for the comment on which we agree. In fact, in Figure.13 we correlated the yearly measured total yield of the main types of crops in the two areas with the ADMOS yearly value. However, both areas are highly irrigated, so that no big differences are found in terms of crop yields in the different years. Therefore, it becomes important and significant to consider the impact of drought on irrigation volumes. We agree that availability also depends on groundwater and the filling availability of reserves, but this would be directly reflected in the volume of water which was effectively used for irrigation. This use also depends on the need for plants and therefore on the weather conditions. Thus, we think that correlating the ADMOS index with the effective water use for irrigation is an effective way of demonstrating that a particular season was drier than another. In Figure.12 a clear correlation is in fact observable.

- .105. Crop yield is also constrained by VPD anomalies sand increases in the atmospheric evaporative demand, particularly under low soil moisture conditions.
  .106. Increase in crop temperature can be also caused by decreased leaf stomatal conductance as consequence of increased VPD.

Answer: Crop yield of course is mainly constrained by soil water availability, VPD or air temperatures. All these aspects are indirectly taken into account by considering in the ADMOS index, the vegetation anomaly and the land surface temperature one, which directly show if crop is under stress (which could be due to water, temperature...).

- .115. Are the different variables following a normal distribution in order to apply this equation?

Answer: we believe that there has been a misunderstanding here. In this case, we are not making any assumption about the normality of the different variables, since we are not comparing them or processing them in any way. We are only determining the sign of the anomaly, i.e., whether the single values are below or above average. The procedure described in equation 1 is necessary only to provide a sign of the variability with respect to the average and to have an overall idea of the oscillations of the variables, each with respect to their mean. Numerous meteorological and drought-detection applications involving anomalies do not require any normality of the variables of interest, as for example done in Blenkinsop and Fowler (2007) and Phillips and McGregor (2001).

1. Blenkinsop, S. and Fowler, H.J. (2007), Changes in European drought characteristics projected by the PRUDENCE regional climate models. Int. J. Climatol., 27: 1595-1610. https://doi.org/10.1002/joc.1538
2. Phillips, I.D. and McGregor, G.R. (1998), The utility of a drought index for assessing the drought hazard in Devon and Cornwall, South West England. Met. Apps, 5: 359-372. https://doi.org/10.1017/S1350482798000899

- .117-120. It is confuse if the authors are using the monthly or daily scales.

Answer: we thank the reviewer for this question, which helps us to clarify this point that has been also raised by other reviewers. The anomalies are computed using the daily data (e.g. when available according to satellite acquisition time) and normalized according to the long-term mean daily values. The only index computed at monthly scale is the SPI-1 as for its definition related to 1 month anomaly. The index at this time scale might be useful in a on-demand irrigation scheme (as the Capitanata case study), to know the evolution of water availability as well as crops conditions and to infer the possible increase or decrease of irrigation water requests from the farmers allowing to better manage the water at Consortium scale at the same time scale of the operating water management system.

- .124. Figure 1 is confuse. It is not clear how the different indices are merged in order to generate the ADMOS. What is the criterion followed to select the thresholds?

Answer: the ADMOS is structured as a "point-system" index. Any time that one of the conditions detailed in the figure is verified, a point (or half point) is deducted. The half points were used to describe milder conditions, with the given anomaly (in absolute value) between 0 and 1, meaning that the value was different from the average, but still it was found within one standard deviation of the average. In joint response to another reviewer, an updated version of fig.1 has been developed (Fig.A), shown below.

[Figure]

Fig.A Improved version of Fig.1 from the manuscript, detailing the ADMOS workflow/flowchart

- .130. Are equations 2 and 3 necessary? I do not think necessary to include the equation of the Pearson's r statistic.

Answer: we wanted to include all the equations and not give anything as known. Nevertheless, we will remove these equations in the new revised manuscript.

- .142. was affected? As the sentence refers to 2012 I think better use the past. Same 143.

Answer: thanks, we will correct these English issues in the new paper version.

- .159. How robust is the calculation of SPI and the other drought indices based only on 20 years of data? e.g.in 168 in is indicated that 13 years of data are used. This will provide very uncertain indices. WMO recommends at least 30 years.

Answer: We agree on this. The SPI-1 index is computed using both ERA-5 data and ground stations network. The ERA-5 data are from 2000 to 2020, while the ground data are available for shorter time periods. So that the precipitation SPI-1 is calculated using both dataset and compared (section 3.1), showing a between the SPI series a RMSE of 0.33 mm and Pearson coefficient of 0.92 for the Capitanata Consortium, and similarly for the Chiese Consortium area with a RMSE of 0.52 mm and a r of 0.97. We will make this clearer in the text.

- Section 22.3. It is very confuse how all these soil moisture indices of different resolution and time span are used together. There is not explanation and justification of why these different soil moisture products are used and what is the advantage of using different datasets if they show low agreement.

Answer: soil moisture products from remote sensing are widely diffused in numerous kinds of applications and provide valuable information on spatial variability, especially over large areas (Babaeian et al., 2019). Nevertheless, as for other remote sensing products, some issues on uncertainties are present. Different works compare these soil moisture products: an example is offered by Cui et al. (2017), who tested SSM data from SMAP, SMOS, AMSR2 and ESA-CCI, among others, obtaining medium-to-high correlations with ground data (ranging from the 0.48 of AMSR2 to 0.89 of SMAP); another by El Hadjj et al. (2018), who compared SMOS, SMAP, ASCAT, and Sentinel-1 SSM products, also employing on-ground measurements and obtaining slightly better correlation results for SMAP (higher than 0.6) than ASCAT (around 0.5) and SMOS (lower than 0.5). A similar comparison was also performed by Paciolla et al. (2020), who contrasted each product with the precipitation occurred in each pixel, computed from ground stations. Medium-level correlations were found between the two, varying heavily from one dataset to the other.

In the image below, we compared the average SM values over the Capitanata consortium (plotted for simplicity only from 2015 to 2017), to provide an idea of the variability between the datasets. In general, all products are showing a similar dynamic, but with quite different values. SMOS and SMAP (L-band products) have similar shapes in the detecting the low SM periods. Another issue between the products is their consistency with water inputs (precipitation and artificial irrigation), as discussed in Paciolla et al. (2020).

[Figure]

Given this heterogeneity, we felt that choosing a single product and limiting out analysis to that product and that product would capture only partially the spectrum of possible values. Thus, we decided that employing the different products all together would lead to an increase in robustness for the whole ADMOS methodology.

1. Cui, C.; Xu, J.; Zeng, J.; Chen, K.-S.; Bai, X.; Lu, H.; Chen, Q.; Zhao, T. Soil Moisture Mapping from Satellites: An Intercomparison of SMAP, SMOS, FY3B, AMSR2, and ESA CCI over Two Dense Network Regions at Different Spatial Scales. Remote Sens. 2017, 10, 33
2. El Hajj, M.; Baghdadi, N.; Zribi, M.; Rodríguez-Fernández, N.J.; Wigneron, J.-P.; Al-Yaari, A.; Al Bitar, A.; Albergelb, C.; Albergel, C. Evaluation of SMOS, SMAP, ASCAT and Sentinel-1 Soil Moisture Products at Sites in Southwestern France. Remote Sens. 2018, 10, 569
3. Paciolla, Nicola, et al. "Irrigation and precipitation hydrological consistency with SMOS, SMAP, ESA-CCI, Copernicus SSM1km, and AMSR-2 remotely sensed soil moisture products." Remote Sensing 12.22 (2020): 3737.

- .214. Why thermal bands are resampled to 100 meters?

Answer: the original LANDSAT8 data are collected directly at 100m, but we resampled them at 30 m to be consistent with the other datasets.

- Figure 3. It is impossible to identify the drought periods according to the SPI based on this plot. I would suggest to be replaced all the plots by time series.
- Figure 4. Same that for precipitation. I do not think it is possible to compare these different datasets based on these plots. The statistics that compare the datasets suggest strong uncertainties and difficulties for comparison. I do not think that the authors are providing realiable combination of the different datasets and, in addition, validation is not provided.
- Same comments are valid for surface temperature and vegetation indices. My impression is that authors have used all the information they have found by different sources, but they have not considered any coherent approach to analyse drought severity, to validate the different products and to stablish uncertainties associated to the datasets. In addition, the information is not showed in a coherent way and it is very difficult to determine the evolution of the anomalies in the different metrics and also to establish comparisons.
- 549: But the remote sensing information is not used in a coherent way considering a careful validation. Several datasets are put together considering different time periods and I cannot find a coherent message by so confuse merging.

Answer: Thanks for the comments, we respond here to the group of the last three comments on a similar topic.

Regarding the plots by time series, we believe that the simple time series plots increase the confusion in the reading and interpretation of data, so that we chose the kind of plot present in the paper on purpose. Below we provided some time series plots of the single variables and, as you can see, it is difficult to detect any seasonality in the values across the different years. The whole thing results poorly comprehensible, whereas in our original plots you can easily compare the values for a given month/period across the different years, for the SPI and the other variables.

[Figure]

[Figure]

[Figure]

[Figure]

[Figure]

Regarding SM, as replied also to other comments, soil moisture products from remote sensing are widely diffused in numerous kinds of applications and provide valuable information on spatial variability, especially over large areas (Babaeian et al., 2019). Also given the heterogeneity and uncertainty of SM products from remote sensing, we felt that choosing the different products all together would lead to an increase in robustness for the whole ADMOS methodology.

Figures 4, 6, 7 and 8 allow the comparison among RS sources, providing a way to understand the possible uncertainties and differences that are present among the various sources of information.

However, we agree that, for the comparison of the different anomalies, we could add a single time plot series with monthly aggregated values. Below we provide an example for the Capitanata dataset, where the SPI negative anomaly of July-August 2011 is followed by a high negative SMAI in August-September and followed by a positive LSTA. Similarly in the summer of 2012, where negative SPI and SMAI are followed by positive LSTA and negative NDVIA. The one for Chiese will also be added in the revised version of the manuscript.

[Figure]

Precisely because these SM data may have uncertainties, but at the same time are widely used, the robustness of the presented approach is based on a combined use of these data. In Table.2 all the products are intercompared, providing evidence of the uncertainty of the different datasets.

If with Validation you mean the validation of remote sensing products we can argue that we did not developed own products but we used already available and validated ones. Moreover, when comparing local sparse ground data with remote sensing ones, the scale and representativeness of both data need to be considered. Instead if for validation you mean of the entire index, to reinforce the strength of the methodology each single anomaly is now correlated with the irrigation volume and crop yield (as shown in one of the previous comments).

- Figures 9 and 10: Based on the uncertainties in the datasets and methods indicated above, the uncertainty in the results described based on these figures are very strong. It is not possible to infer on which dataset (e.g. soil moisture, surface temperature and vegetation index) this plot is generated.

Answer: Thanks for the comment, which helps us to correct this oversight in specifying the dataset used for Figure.9 and 10. Actually, in fact, the datasets on which the different drought conditions are based will be specified: SPI ERA-5, SMA ESA-CCI combined, LSTA and NDVIA MODIS.

- .445. I cannot identify how the different products are combined in order to generate the ADMOS and it is very confuse the use of different data products at the same time and in an independent way.

Answer: as described in the methodology section, each product is analysed in terms of anomaly and, if the conditions detailed in Figure 1 are met, the corresponding ADMOS level is assigned to each day. We developed a more clear version of fig.1 to better convey its message (fig.A here below).

[Figure]

Fig.A Improved version of Fig.1 from the manuscript, detailing the ADMOS workflow/flowchart

- .545: I agree that different indices are compared, but this is not done in this study. There is not validation of different metrics and selection of most suitable according to empirical information.

Answer: As replied also to the previous comment, we agree with you that such a comparison would be a valuable addition to the workflow of the whole paper. We will add these comparisons to the final version of the work.

---

## Author Comment (AC4)

**RC3**

I believe the topic is relevant, and I think the paper might merit the chance to be published eventually, but it definitely needs major revisions.

Some general comments

- I believe the purpose of the paper is lost in divagations due to the reporting style and contents. The ADMOS indicator results should be more explained, in particular regarding its capacity to reproduce impacts on yield or its covariation with irrigation. Since the indicator values and thresholds are arbitrary, it is essential to see if such thresholds are capable of marking when impacts on crop yields are to be expected- which seems not the case, or when irrigation inputs are necessary, and in what order of magnitude. It is not clear to me in the end it is possible to use ADMOS to recommend better water management for agriculture.

Answer: thanks for this comment, on which we agree with you that such a comparison would be a valuable addition to the workflow of the whole paper, to understand the effectiveness of the ADMOS index in respect to single anomalies indicator for water management. ADMOS is conceived as a monitoring tool, to be employed in order to follow more closely the evolution of possible drought dynamics. The index might be useful in a on-demand irrigation scheme (as the Capitanata case study), to know the evolution of water availability as well as crops conditions and to infer the possible increase or decrease of irrigation water requests from the farmers allowing to better manage the water at Consortium scale.

- Since several indices have been calculated to compose the ADMOS, I was expecting a separate comparison between them and the irrigation and yields. Maybe them separately have better prediction capacity than the ADMOS itself, but it was not showed. The conclusions say that you prove that "droughts cannot be described by one single indicator but there is the need first to select the correct physical index for detecting a drought type and secondly to use different drought indices to identify specific conditions",  but I don´t see how you compare the ADMOS predictive capacity (for irrigation or crop yields) and the predictive capacity of all your anomalies series for P, SM, temperature or VI.

Answer: we agree with you that such a comparison would be a valuable addition to the workflow of the whole paper. We will add these comparisons to the final version of the work but, in the meanwhile, we attach here some plots, conceptually similar to those in Figure.12 and 13, but featuring the single anomalies (average value in the crops season) instead of the ADMOS, over the Capitanata test case. The strength of ADMOS is particularly relevant when you compare the single anomalies with the cumulated irrigation volume, where you obtain a positive dependency (at higher SPI values you would expect to use lower irrigation water). Also with SMA from remote sensing uncertainties in the correlation between SM dynamic and irrigation events. Moreover, comparing the determination coefficients (R2) with those from the figures in the manuscript, the correlation between cumulated volume from the aqueduct and single anomaly is everywhere weaker with respect to the ADMOS. This difference is even wider when considering all water inputs (aqueduct and natural rainfall), with the correlations falling below 0.20. Finally, when looking at the final yield (third figure, to be contrasted with fig.13 from the manuscript), a similar behaviour is observed if considering the ADMOS or single anomalies.

[Figure]

- Also, it is very difficult to understand the spatial and temporal aggregation scales for the indices and the ADMOS indicator itself. It seems it is calculated daily, but then the yields are annual for the entire consortium, not sure though. What about the irrigation values? They don´t even appear in the data list. How often are them recorded? How are them aggregated over time?

Answer: we do agree that bigger clarity is required in the presentation of these topics, and will improve the relative excerpts of the article. To answer your doubts, ADMOS is computed daily, and the values that are compared to the consortium-level irrigation water volumes are the annual cumulated ADMOS values, summed up day by day to identify a value representative of the whole year. For what concerns the irrigation values, they are registered by the irrigation consortia day by day (for the Chiese case, their distribution is planned in advance, while for the Capitanata, the on-demand irrigation means that the volumes are strictly measured to determine the amount to pay for each farmer). The annual value shown in the last figure is obtained by summing the whole annual time series.

- On the other hand, the paper devotes too much space (6 full pages, and 4 chapters, 3.1 to 3.4) to debate the differences between each product (RMSE, r) in each of the variables, when it is not the essential result and could be solved with a summary table and a paragraph of explanation.

Answer: Yes, thanks we agree with your comment. We then plan to reduce the number of figures, by deleting Figure 7. which is indeed summarized in table.3, as well we will delete Figure.5 and we will discuss the monthly correlation variations in the text.

- I have doubts if- from a statistical point of view- it is recommendable to accumulate an indicator (ADMOS) whose values are categories and not quantifications. For example, accumulating two time steps with ADMOS -1 is -2, but it is not necessarily equivalent to another time step with -2, and still they are added up. That might explain that in summer there are peaks in the accumulation. It should be better justified why it is computed like this.

Answer: thanks for your observation. It is true that, being categories, the single values of the ADMOS index seem ill-suited to a cumulation. However, we feel that the main aim of ADMOS is not characterizing the single drought (whether it is developing as a precipitation, soil moisture, thermal or vegetation drought), but simply provide an idea of the total magnitude of it.

- In any case, does a value of cumulative ADMOS at certain point means that at that moment a certain irrigation volume should be applied? In the conclusions, the papers says "This ADMOS might help irrigation districts managers and farmers to activate the preventive protection actions to try to avoid water volume and crop yield losses.", but after reading the study I don´t really see how, it would merit an explanation.

Answer: Thanks for the comment, yes as you can see in the following figure the increase of cumulated volume really used for irrigation during the season as a clear correspondence with the negative increase of the ADMOS index. So that it might be useful for monitoring water requests.

[Figure]

- Also, the potential lags / delays are not taken into consideration for comparing the evolution of the different variables. Only SPI1 is calculated and confronted with the situation at the same time in other variables, presumably at the daily level, but not sure you can capture the propagation of the anomalies like this. For example, time-steps with very high negative anomalies in SM or Vegetation Index can be concealed by the fact that that day in particular it rained a lot, the ADMOS would show "surplus of water", but the system has not recovered yet. There is not a way to know if ADMOS then marks real issues in terms of irrigation needs or yield losses at the daily level. With more granular data on these two impacts, tests could be made.
- More generally, I do not really see how the ADMOS is helping identify the main agricultural drought problems in the pilots used. More examples, maybe using a particular event, would help strengthening that point.

Answer: we provided here an example that could also be added to the main manuscript as an operative demonstration of how ADMOS could be helpful in tackling drought monitoring/identification issues. In the sub-figure below, the four analysed anomalies (SPI, SMA, LSTA, VISA) are shown during the 2012 agricultural season. By looking at their evolution, it would seem that a negative anomaly in precipitation (blue line) during May was followed by a negative one in soil moisture (brown) and vegetation index (green) and a positive one in LST (red) during June and July. At the time, SPI seems to have recovered at milder values. What ADMOS manages to do is merge all this information together in one single index, providing a complete overview of the status of the entire soil-plant ecosystem. Finally, for what concerns the tricky "surplus of water" question, we should have clarified that positive ADMOS values were never featured in the computation of the cumulated value. This was done (i) because ADMOS is not meant to identify water-abundance conditions and (ii) because single heavy-rain days could "nullify" the build-up of the previous days with a positive value (which is exactly

the case you referred to). In order to avoid further confusion, we removed the "surplus of water" conditions (positive SPI-1) from the main methodological table.

[Figure]

- Last, on a different note, the used references do not seem always the most relevant to justify the points the authors make, sometimes it is just general drought literature, not even focused on reviewing similar efforts.

Answer: thank you for the observation, we received a similar comment from another reviewer. We have already updated some of the references during this revision process and will revise the references entirely for the final version of the manuscript.

Some specific comments

- 450- "All the curves have a common trend: from the begin of the year till March the curves have gentle slopes, then from March to October they are very steep due to drought conditions, and at the end of the year they return flat." I think this reveals that the drought indicator is more marking stress than drought, as it points to systematic intensification in summer

Answer: we will add a comment on this in the text, but the ADMOS index is capturing also winter drought periods, not only summer. This is now clearly visible from the Figure we added here in a previous comment on the cumulated ADMOS and volumes for Capitanata Consortium, where a steep increase in ADMOS is visible during November of 2011 and 2012.

- Figure 9 and 10. "Synchronicity among the different variables' anomalies in… "- I don´t think the color code and the graphs are easy to read and interpret.

Answer: the plots will be improved to eliminate any possible ambiguity.

- 515- "Following the principles of CAP to improve irrigation management"- Spell CAP.

Answer: it is the Common Agricultural Policy, will be added to the manuscript final version.

- 540- "This methodology improves the traditional analysis, which are generally analysed by considering only soil moisture anomalies". Many drought analyses rely on other variables.

Answer: we agree with your comment, we will rephrase specifying that ADMOS is more tailored for irrigated agricultural area

- The writing is confusing in many parts and there are several typos or incongruences, it needs a language revision.

Answer: the writing will be revised all over the manuscript for the final version.